# CoarsenConf: Equivariant Coarsening with Aggregated Attention for Molecular Conformer Generation

## Abstract

Molecular conformer generation (MCG) is an important task in cheminformatics and drug discovery. The ability to efficiently generate low-energy 3D structures can avoid expensive quantum mechanical simulations, leading to accelerated virtual screenings and enhanced structural exploration. Several generative models have been developed for MCG, but many struggle to consistently produce high-quality conformers. To address these issues, we introduce CoarsenConf, which coarse-grains molecular graphs based on torsional angles and integrates them into an SE(3)-equivariant hierarchical variational autoencoder. Through equivariant coarse-graining, we aggregate the fine-grained atomic coordinates of subgraphs connected via rotatable bonds, creating a variable-length coarse-grained latent representation. Our model uses a novel aggregated attention mechanism to restore fine-grained coordinates from the coarse-grained latent representation, enabling efficient generation of accurate conformers. Furthermore, we evaluate the chemical and biochemical quality of our generated conformers on multiple downstream applications, including property prediction and oracle-based protein docking. Overall, CoarsenConf generates more accurate conformer ensembles compared to prior generative models.

## 1 Introduction

Molecular conformer generation (MCG) is a fundamental task in computational chemistry. The objective is to predict stable low-energy 3D molecular structures, known as conformers. Accurate molecular conformations are crucial for various applications that depend on precise spatial and geometric qualities, including drug discovery and protein docking. For MCG, traditional physics-based methods present a trade-off between speed and accuracy. Quantum mechanical methods are more accurate but computationally slow, while stochastic cheminformatics-based methods like RDKit ETKDG (Riniker and Landrum, 2015) provide more efficient but less accurate results. As the difficulty of computing low-energy structures increases with the number of atoms and rotatable bonds in a molecule, there has been interest in developing machine learning (ML) methods to generate accurate conformer predictions efficiently.

Existing generative MCG ML models can be broadly classified based on two primary criteria: (1) the choice of the architecture to model the distribution of low-energy conformers and (2) the manner in which they incorporate geometric information to model 3D conformers. A majority of these methods utilize a single geometric representation, such as distance matrices, atom coordinates, or torsional angles, and pair them with various probabilistic modeling techniques, including variational autoencoders (VAEs) and diffusion models. A recurring limitation of these prior approaches is that they are restricted to modeling only one specific aspect of geometric information *i.e.*, coordinates or angles. By restricting themselves to either a pure coordinate or angle space, they often fail to fully leverage the full geometric information inherent to the problem. Consequently, in this work, we introduce a more flexible molecular latent representation that seamlessly integrates multiple desired geometric modalities, thereby enhancing the overall accuracy and versatility of conformer generation.

To create this flexible representation, we use a process known as coarse-graining to distill molecular information into a simplified, coarse-grained (CG) representation based on specific coarsening criteria. This is analogous to previous 2D fragment-based generative techniques (Chen et al., 2021).

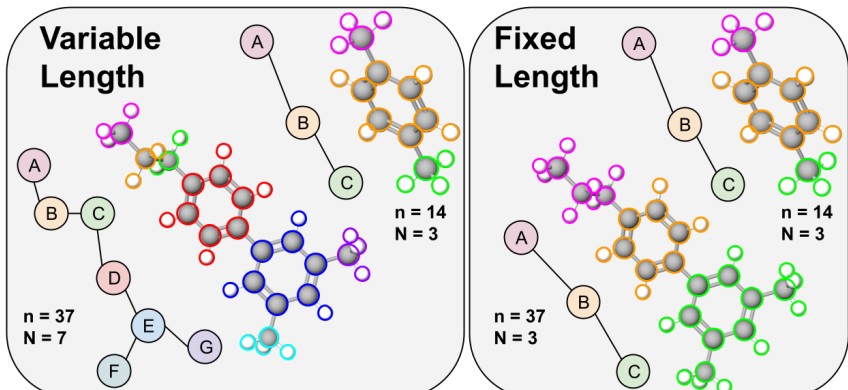

Figure 1: **Learning flexible coarse-grained representations.** CoarsenConf is the first model to employ variable-length (on left) coarse-graining. Each input molecule ($n$ fine-grained (FG) atoms) can be represented by a different number of coarse-grained (CG) nodes $N$, thus accommodating diverse molecular sizes. In contrast, prior approaches rely on fixed-length (on right) coarse-graining, thereby forcing all molecules to possess the same number of CG nodes. Variable-length coarse-graining enhances the model's ability to create better learned representations across molecules of different sizes and geometries. The molecules on the left are coarsened along torsional angles.

In these scenarios, the fragmentation of molecules into prominent substructures led to significant enhancements in representation learning and generative tasks. Coarse-graining has seen success in related ML applications like molecular dynamics and protein modeling (Husic et al., 2020; Kmiecik et al., 2016). However, these ML approaches have primarily explored rigid coarsening criteria, such as distance-based nearest neighbor methods, which represent all inputs with a fixed granularity or number of beads (*i.e.* CG nodes, $N$). ML models that use fixed-length coarse-graining often necessitate $N$ be fixed for all input molecules (Yang and Gomez-Bombarelli, 2023). This approach may not be suitable for all scenarios, especially when navigating multi-modal datasets of drug-like molecules of varying sizes (multiple $N$), which better reflects real-world conditions.

To address the limitations of fixed-length CG representations, we introduce *Aggregated Attention* for variable-length coarse-graining (see Fig. 1). This methodology allows a single latent representation to accommodate molecules with different numbers of fine-grained (FG) atoms and CG beads. The inherent flexibility of the attention mechanism allows input molecules to be fragmented along torsion angles, enabling the modeling of interatomic distances, 3D atom coordinates, and torsion angles in an equivariant manner, regardless of the molecule's shape or size. Through Aggregated Attention, we also harness information from the entire learned representation, unlike preceding approaches that restrict the backmapping (from CG back to FG) to a subset (Wang et al., 2022). The adaptability of our learned variable-length representations enables more accurate generation.

Our full innovations are encapsulated in CoarsenConf, an SE(3)-equivariant hierarchical VAE. CoarsenConf aggregates information from fine-grained atomic coordinates to create a flexible subgraph-level representation, improving the accuracy of conformer generation. Unlike prior MCG methods, CoarsenConf generates low-energy conformers with the ability to model atomic coordinates (FG and CG), distances, and torsion angles directly via variable-length coarse-graining.

Our main contributions are as follows:
- We present CoarsenConf, a novel conditional hierarchical VAE. CoarsenConf learns a coarse-grained or subgraph-level latent distribution for SE(3)-equivariant conformer generation. To our knowledge, this is the first method to use coarse-graining in the context of MCG.

- CoarsenConf is the first model capable of handling variable-length coarse-to-fine generation without a molecule fragment vocabulary, using an *Aggregated Attention* strategy. Coarsen-Conf employs a single flexible variable-length node-level latent representation that can uniquely represent molecules of any size with any number of coarse-grained nodes. Furthermore, variable-length coarse-graining circumvents having to train separate generative models for each number of CG beads to represent the same molecular dataset accurately (100+ models for MCG), which is a limitation of fixed-length methods (Yang and Gomez-Bombarelli, 2023).

- We predominantly outperform prior methods on GEOM-QM9 and GEOM-DRUGS for RMSD precision and property prediction benchmarks (Axelrod and Gómez-Bombarelli, 2022). We also produce a lower overall RMSD distribution across all conformers, achieving this with an order of magnitude less training time compared to prior methods.

- We evaluate CoarsenConf on multiple downstream applications to assess the chemical and biochemical quality of our generated conformers, including oracle-based protein docking (Huang et al., 2021) (the affinity of generated conformers to bind to specific protein pockets) under both flexible and rigid conformational energy minimizations. Despite lacking prior knowledge about the protein or the downstream task, CoarsenConf generates significantly better binding ligand conformers for known protein binding sites when compared to prior MCG methods for both oracle scenarios.

## 2   BACKGROUND

**Notations.**   We represent each molecule as a graph $G = (\mathcal{V}, \mathcal{E})$, where $\mathcal{V}$ is the set of vertices representing atoms and $\mathcal{E}$ is the set of edges representing inter-atomic bonds. Each node $v$ in $\mathcal{V}$ describes the chosen atomic features such as element type, atomic charge, and hybridization state. Each edge $e_{uv}$ in $\mathcal{E}$ describes the corresponding chemical bond connecting $u$ and $v$, and is labeled with its bond type. Following Simm and Hernandez-Lobato (2020), each molecular graph is expanded to incorporate auxiliary edges connecting all atoms within a 4Å radius to enhance long-range interactions in message passing. The spatial position of each atom in $\mathcal{V}$ is represented by a 3D coordinate vector $\boldsymbol{r} \in \mathbb{R}^3$, such that the full molecule conformation is represented by the matrix $\boldsymbol{X} \in \mathbb{R}^{|\mathcal{V}| \times 3}$.

**Problem Definition.**   *Molecular conformation generation* (MCG) is a conditional generative process that aims to model the conditional distribution of 3D molecular conformations $\boldsymbol{X}$, given the 2D molecule graph $G$, *i.e.*, $p(\boldsymbol{X}|G)$. While prior works have shown some success with learning 3D conformations starting with only the 2D information, many require complex and compute-intensive architectures (Zhu et al., 2022; Zhou et al., 2023). Recently, Jing et al. (2022) demonstrated good performance on the GEOM-DRUGS dataset by priming the method with easy-to-obtain 3D approximations via RDKit ETKDG (Riniker and Landrum, 2015). Jing et al. (2022) showed that RDKit is highly effective at generating conformations with correct bond distances and, as a result, can constrain the problem to a diffusion process over only torsion angles.

We thus formalize MCG as modeling the conditional distribution $p(\boldsymbol{X}|\mathcal{R})$, where $\mathcal{R}$ is the RDKit generated atomic coordinates. This is functionally the same underlying distribution as $p(\boldsymbol{X}|G)$, as we use RDKit as a building block to provide an approximation starting from only 2D information, $\mathcal{R} = \text{RDKit ETKDG}(G)$. We will show that more robust conformers are generated by conditioning on approximations without imposing explicit angular and distance constraints.

**Classical Methods for Conformer Generation.**   A molecular conformer refers to the collection of 3D structures that are energetically favorable and correspond to local minima of the potential energy surface. CREST (Pracht et al., 2020) uses semi-empirical tight-binding density functional theory (DFT) for energy calculations, which, while computationally less expensive than ab-initio quantum mechanical (QM) methods, still requires approximately 90 core hours per drug-like molecule (Axelrod and Gómez-Bombarelli, 2022). Though CREST was used to generate the "ground truth" GEOM dataset, it is too slow for downstream applications such as high-throughput virtual screening.

Cheminformatics methods, such as RDKit ETKDG, are commonly used to quickly generate approximate low-energy conformations of molecules. These methods are less accurate than QM methods due to the sparse coverage of the conformational space resulting from stochastic sampling. Additionally, force field optimizations are inherently less accurate than the above QM methods. RDKit ETKDG employs a genetic algorithm for Distance Geometry optimization that can be enhanced with a molecular mechanics force field optimization (MMFF).

**Deep Learning Methods for Conformer Generation.**   Several probabilistic deep learning methods for MCG have been developed (Anstine and Isayev, 2023), such as variational autoencoders in CVGAE (Mansimov et al., 2019) and ConfVAE (Xu et al., 2021b), normalizing flows in CGCF (Xu et al., 2021a), score-based generative models in ConfGF (Shi et al., 2021) and DGSM (Luo et al., 2021), and diffusion models in GeoDiff (Xu et al., 2022) and Torsional Diffusion (Jing et al., 2022). GraphDG (Simm and Hernandez-Lobato, 2020) forgoes modeling coordinates and angles, relying solely on distance geometry. DMCG (Zhu et al., 2022) and Uni-Mol (Zhou et al., 2023) present

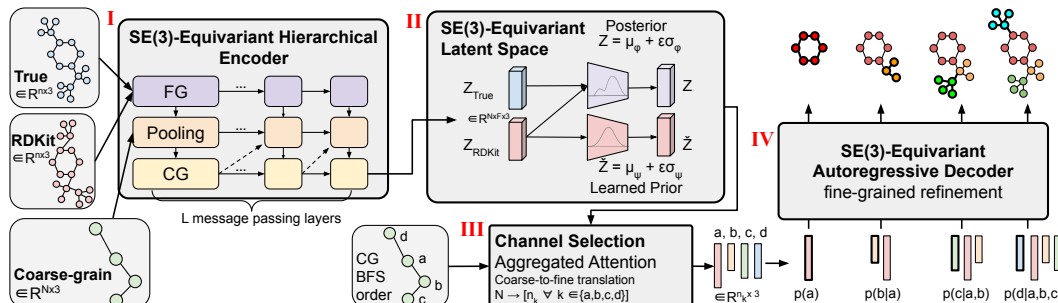

Figure 2: **CoarsenConf architecture. (I)** The encoder $q_\phi(z|X, \mathcal{R})$ takes fine-grained (FG) ground truth conformer $X$, RDKit approximate conformer $\mathcal{R}$, and coarse-grained (CG) conformer $\mathcal{C}$ as inputs (derived from $X$ and the predefined CG strategy), and outputs a variable-length equivariant CG representation via equivariant message passing and point convolutions. **(II)** Equivariant MLPs are applied to learn the mean and log variance of both the posterior and prior distributions. **(III)** The posterior (training) or prior (inference) is sampled and fed into the Channel Selection module, where an attention layer is used to learn the optimal mapping from CG to FG structure. **(IV)** Given the FG latent vector and the RDKit approximation, the decoder $p_\theta(X|\mathcal{R}, z)$ learns to recover the low-energy FG structure through autoregressive equivariant message passing. The entire model can be trained end-to-end by optimizing the KL divergence of latent distributions and reconstruction error.

examples of effective large models, the first mimicking the architecture of AlphaFold (Jumper et al., 2021) and the second using large-scale SE(3)-equivariant transformer pre-training.

**Molecular Coarse-graining.** Molecular coarse-graining refers to the simplification of a molecule representation by grouping the fine-grained (FG) atoms in the original structure into individual coarse-grained (CG) beads with a rule-based mapping.[1] Coarse-graining has been widely utilized in protein design (Kmiecik et al., 2016) and molecular dynamics (Gkeka et al., 2020), and analogously fragment-level or subgraph-level generation has proven to be highly valuable in diverse 2D molecule design tasks (Chen et al., 2021). Breaking down generative problems into smaller pieces can be applied to several 3D molecule tasks. We note many prior fragment-based generative strategies rely on adding fixed fragments from a predefined vocabulary. For instance, CGVAE (Wang et al., 2022) learns a latent distribution to back map or restore FG coordinates from a fixed number of CG beads effectively. We also note that various coarse-graining strategies exist (Jin et al., 2022a; Arts et al., 2023; Husic et al., 2020; Chennakesavalu et al., 2023), and many require the ability to represent inputs with a non-fixed granularity. To handle this, CoarsenConf uses a flexible variable-length CG representation that is compatible with all coarse-graining techniques.

We include an extensive discussion on further relevant work in Appendix §A. We discuss prominent 2D and 3D equivariant autoregressive molecular generation, protein docking, and structure-based drug discovery (SBDD) techniques, as well as our formal definition of SE(3)-equivariance.

## 3 METHODS

**Coarse-graining Procedure.** We first define a rotatable bond as any single bond between two non-terminal atoms, excluding amides and conjugated double bonds, where the torsion angle is the angle of rotation around the central bond. Formally, the torsion angle $\tau_{abcd}$ is defined about bond $(b, c) \in \mathcal{E}$ where (a, b) are a choice of reference neighbors s.t $a \in \mathcal{N}(b) \setminus c$ and $d \in \mathcal{N}(c) \setminus b$.

We coarsen molecules into a single fragment or bead for each connected component, resulting from severing all rotatable bonds. This choice in CG procedure implicitly forces the model to learn over torsion angles, as well as atomic coordinates and inter-atomic distances. We found that using a more physically constrained definition of torsional angles, as defined by Ganea et al. (2021), in the CG procedure led to a significant increase in performance compared to that used in Jing et al. (2022). This is because the latter allows rotations around double and triple bonds, while the former does not. An example of the coarse-graining procedure is in Fig. 1. For formal definitions, see Appendix §C.

**Learning Framework.** CoarsenConf is a conditional generative model that learns $p(X|\mathcal{R})$ where $X$ is the low-energy 3D conformation, and $\mathcal{R}$ is the RDKit approximate conformation. Specifically,

---

[1]We use the terms CG graph nodes and beads interchangeably.

we optimize $p(X|\mathcal{R})$ by maximizing its variational lower bound with an approximate conditional posterior distribution $q_\phi(z|X, \mathcal{R})$ and learned conditional prior $p_\psi(z|\mathcal{R})$:

$$\log p(X|\mathcal{R}) \geq \underbrace{\mathbb{E}_{q_\phi(z|X,\mathcal{R})} \log p_\theta(X|\mathcal{R}, z)}_{\mathcal{L}_{\text{reconstruction}}} + \underbrace{\mathbb{E}_{q_\phi(z|X,\mathcal{R})} \log \frac{p_\psi(z|\mathcal{R})}{q_\phi(z|X, \mathcal{R})}}_{\mathcal{L}_{\text{latent regularization}}} + \mathcal{L}_{\text{auxiliary}}, \tag{1}$$

where $q_\phi(z|X, \mathcal{R})$ is the hierarchical equivariant encoder model, $p_\theta(X|\mathcal{R}, z)$ is the equivariant decoder model to recover $X$ from $\mathcal{R}$ and $z$, and $p_\psi(z|\mathcal{R})$ is the learned prior distribution. The reconstruction loss, $\mathcal{L}_{\text{recon.}}$, is implemented as $\text{MSE}(\mathcal{A}(X_{true}, X_{model}), X_{model})$, where $\mathcal{A}$ is the Kabsch alignment function that provides an optimal rotation matrix and translation vector to minimize the mean squared error (MSE) (Kabsch, 1993). The second term, $\mathcal{L}_{\text{reg.}}$, can be viewed as a regularization over the latent space and is implemented as $\beta D_{KL}(q_\phi(z|X, \mathcal{R}) \parallel p_\psi(z|\mathcal{R}))$ (Higgins et al., 2017). More details on the geometric auxiliary loss function are in Appendix §B.

**Encoder Architecture.** CoarsenConf's encoder, shown in Fig. 2(I), operates over SE(3)-invariant atom features $h \in R^{n \times D}$, and SE(3)-equivariant atomistic coordinates $x \in R^{n \times 3}$. A single encoder layer is composed of three modules: fine-grained, pooling, and coarse-grained. Full equations for each module can be found in Appendix §D.1, §D.2, §D.3, respectively. The encoder module takes in $x$ and $h$ from the ground truth and RDKit conformer and creates coarse-grained latent representations for each, $Z$ and $\tilde{Z} \in R^{N \times F \times 3}$, where $N$ is the number of CG beads, and $F$ is the latent dimensions.

**Equivariant Latent Space.** As $Z$ holds a mixture of equivariant spatial information, we maintain equivariance through the reparametrization trick of the VAE (Fig. 2(II)). Specifically, we define the posterior and prior means ($\boldsymbol{\mu}_\phi$, $\boldsymbol{\mu}_\psi$) and standard deviations ($\boldsymbol{\sigma}_\phi$, $\boldsymbol{\sigma}_\psi$), as follows:

$$\begin{aligned} \text{Posterior}: \boldsymbol{\mu}_\phi &= \text{VN-MLP}(Z, \tilde{Z}), \quad \log(\boldsymbol{\sigma}_\phi^2) = \text{MLP}(Z, \tilde{Z}), \\ \text{Prior}: \boldsymbol{\mu}_\psi &= \text{VN-MLP}(\tilde{Z}), \qquad \log(\boldsymbol{\sigma}_\psi^2) = \text{MLP}(\tilde{Z}). \end{aligned} \tag{2}$$

We use an invariant MLP to learn the variance and apply it to the x, y, and z directions to enforce equivariance. We note that the conditional posterior is parameterized with both the ground truth and RDKit approximation, whereas the learned conditional prior only uses the RDKit.

**Decoder Architecture: Channel Selection.** We sample from the learned posterior (training) and learned prior (inference) to get $Z = \mu + \epsilon\sigma$, where $\epsilon$ is noise sampled from a standard Gaussian distribution as the input to the decoder. Given $Z$ is still in CG space, we need to perform variable-length backmapping to convert back to FG space so that we can further refine the atom coordinates to generate the low energy conformer. The variable-length aspect is crucial because every molecule can be coarsened into a different number of beads, and there is no explicit limit to the number of atoms a single bead can represent. Unlike CGVAE (Wang et al., 2022), which requires training a separate model for each choice in CG granularity $N$, CoarsenConf is capable of reconstructing FG coordinates from any $N$ (illustrated in Fig. 2(III) and Fig. 1).

CGVAE defines the process of channel selection (CS) as selecting the top-$k$ latent channels, where $k$ is the number of atoms in a CG bead of interest. Instead of discarding all learned information in the remaining $F - k$ channels in the latent representation, we use a novel aggregated attention mechanism. This mechanism learns the optimal mixing of channels to reconstruct the FG coordinates, and is illustrated in Fig. 3. The attention operation allows us to actively query our latent representation for the number of atoms we need, and draw upon similarities to the learned RDKit approximation that has been distilled into the latent space through the hierarchical encoding process. Channel selection translates the CG latent tensor $Z \in R^{N \times F \times 3}$ into FG coordinates $x_{CS} \in R^{n \times 3}$.

**Decoder Architecture: Coordinate Refinement.** Once channel selection is complete, we have effectively translated the variable-length CG representation back into the desired FG form. From here, we explore two methods of decoding that use the same underlying architecture. $x_{CS}$ can be passed in a single step through the decoder or can be grouped into its corresponding CG beads, but left in FG coordinates to do a bead-wise autoregressive (AR) generation of final coordinates (Fig. 2(IV)). CoarsenConf is the first MCG method to explore AR generation. Unlike prior 3D AR methods, CoarsenConf does not use a pre-calculated molecular fragment vocabulary, and instead conditions directly on a learned mixture of previously generated 3D coordinates and invariant atom features.

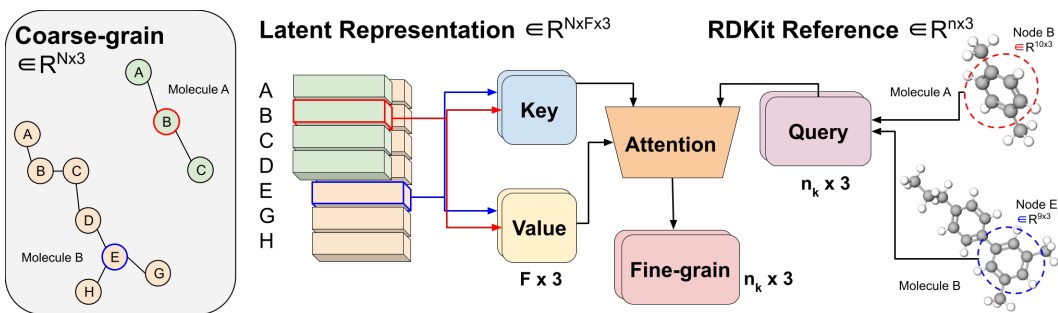

Figure 3: **Variable-length coarse-to-fine backmapping via Aggregated Attention.** The highlighted latent beads of two independent molecules are attended to by the respective fine-grained queries in a batched manner (see red and blue), to generate FG coordinates in the desired shape (matching input queries on right). The single-head attention operation uses the latent vectors of each CG bead $Z_k \in R^{F \times 3}$ for each molecule as the keys and values, with an embedding dimension of 3 to match the x, y, z coordinates. The query vectors are the FG subset of the respective RDKit conformers, corresponding to each CG bead $\in R^{n_k \times 3}$. We know a priori how many FG atoms correspond to a certain CG bead ($n_k$). Aggregated Attention learns the optimal blending of CG features for FG reconstruction by aggregating 3D segments of FG information to form our latent query.

The decoder architecture is similar to the EGNN-based FG module in the encoder, but has one key difference. Instead of learning raw atom coordinates, we learn to predict the difference between the RDKit reference and ground truth conformations. As the goal of MCG is to model the conditional distribution $p(X|\mathcal{R})$, we simplify the learning objective by setting $X = \mathcal{R} + \Delta X$, and learn the optimal distortion $\Delta X$ from the RDKit approximation. The simplification follows, as we have ensured $\mathcal{L}_{\text{recon.}}$ is no worse than that of the RDKit approximation, which is trivial to obtain, compared to the cost of model training and inference.

See Appendix §E for formal discussions of the decoder architecture and message-passing equations.

## 4 EXPERIMENTS

We evaluate MCG on RMSD spatial accuracy (§4.1), property prediction (§4.1), and biochemical quality through flexible (§4.3) and rigid (§4.4) oracle-based protein docking. We include the following models for comparison with CoarsenConf, which are previous MCG methods that use the same train/test split: Torsional Diffusion (TD) (Jing et al., 2022), GeoMol (GM) (Ganea et al., 2021), and when possible, GeoDiff (GD) (Xu et al., 2022). For our model configuration and compute resource breakdown, see Appendix §F. The details of CoarsenConf's initial RDKit structures, as well as how CoarsenConf learns to avoid the distribution shift found in TD, can be found in Appendix §G.

### 4.1 GEOM BENCHMARKS: 3D COORDINATE RMSD

We use the GEOM dataset (Axelrod and Gómez-Bombarelli, 2022), consisting of QM9 (average 11 atoms) and DRUGS (average 44 atoms), to train and evaluate our model. We use the same train/val/test splits from Ganea et al. (2021) (QM9: 106586/13323/1000 and DRUGS: 243473/30433/1000).

**Problem setup.** We report the average minimum RMSD (AMR) between ground truth and generated conformers, and Coverage for Recall and Precision. Coverage is defined as the percentage of conformers with a minimum error under a specified AMR threshold. Recall matches each ground truth conformer to its closest generated structure, and Precision measures the overall spatial accuracy of the generated conformers. Following Jing et al. (2022), we generate two times the number of ground truth conformers for each molecule. More formally, for $K = 2L$, let $\{C_l^*\}_{l \in [1,L]}$ and $\{C_k\}_{k \in [1,K]}$ respectively be the sets of ground truth and generated conformers:

$$\text{COV-Precision} := \frac{1}{K} \left| \{k \in [1..K] : \min_{l \in [1..L]} \text{RMSD}(C_k, C_l^*) < \delta\} \right|,$$

$$\text{AMR-Precision} := \frac{1}{K} \sum_{k \in [1..K]} \min_{l \in [1..L]} \text{RMSD}(C_k, C_l^*),$$

(3)

Table 1: Quality of ML generated conformer ensembles for the GEOM-QM9 ($\delta = 0.5$Å) and GEOM-DRUGS ($\delta = 0.75$Å) test set in terms of Coverage (%) and Average RMSD (Å) Precision. Bolded results are the best, and the underlined results are second best. See Appendix §I- §J for more details.

| Method | QM9-Precision | | | | DRUGS-Precision | | | |
| | Coverage ↑ | | AMR ↓ | | Coverage ↑ | | AMR ↓ | |
| | Mean | Med | Mean | Med | Mean | Med | Mean | Med |
| --- | --- | --- | --- | --- | --- | --- | --- | --- |
| GeoDiff | 50.0 | 33.5 | 0.524 | 0.510 | 23.7 | 13.0 | 1.131 | 1.083 |
| GeoMol | 75.9 | **100.0** | 0.262 | 0.233 | 40.5 | 33.5 | 0.919 | 0.842 |
| Torsional Diffusion ($\ell = 2$) | 78.4 | **100.0** | 0.222 | 0.197 | **52.1** | **53.7** | **0.770** | 0.720 |
| CoarsenConf-OT | **80.2** | **100.0** | **0.149** | **0.107** | 52.0 | 52.1 | 0.836 | **0.694** |

where $\delta$ is the coverage threshold. The recall metrics are obtained by swapping ground truth and generated conformers. We also report the full RMSD error distributions, as the AMR only gives a small snapshot into overall error behavior. See Appendix §J for more baselines.

**Advantages and limitations of RMSD-based metrics.** 3D coordinate-based metrics are commonly used to evaluate ML methods because these models are typically trained using spatial error-based loss functions (*i.e.*, MSE). However, for domain scientists, these metrics can be somewhat challenging to interpret. Low spatial error (measured by RMSD) is not directly informative of free energy, the primary quantity of interest to scientists (Spotte-Smith et al., 2021; Taylor et al., 2023). Additionally, current spatial benchmarks can be categorized into two distinct types: precision and recall. Each of these metrics comes with its own advantages and limitations.

(1) Precision measures the generation accuracy. It tells us if each generated conformer is close to *any* one of the given ground truth structures, but it does not tell us if we have generated the lowest energy structure, which is the most important at a standardized temperature of 0 K. At industrial temperatures, the full distribution of generated conformers is more important than the ability to generate a single ground truth conformer (for the full RMSD error distribution, see Fig. 4).

(2) Recall compares each ground truth to its closest generated conformer. However, in many applications, we are only concerned with obtaining the lowest energy conformer, not all feasible ones. Furthermore, Recall is severely biased by the number of generated conformers for each molecule. Results worsen by up to ~60% for all models when we move from the previously set standard sampling budget for each molecule of $2L$ to min(L/32, 1), where $L$ is the number of ground truth conformers (see Appendix Fig. 7). As we sample more molecules, we greatly influence the chance of reducing the AMR-Recall. Because of this dependency on the number of samples, we focus on the Precision metrics, which were consistent across all tested sample size budgets. We note that $L$ is 104 on average for GEOM-DRUGS (Mansimov et al., 2019), so even $L/32$ is a reasonable number.

**Results.** In Tab. 1, we outperform all models on QM9, and yield competitive results with TD on DRUGS when using an optimal transport (OT) loss (see Appendix §I- §J for more details). Coarsen-Conf also achieves the lowest overall error distribution, as seen in Fig. 4. CoarsenConf-OT uses an OT loss with the same decoder architecture as in Fig. 2, but is no longer autoregressive (see Appendix Eq. 11 for a formal definition). We also see in Appendix Fig. 7 that the recall is heavily dependent on the sampling budget, as performance gaps shrink from ~50% to ~5%. CoarsenConf-OT was trained for 15 hours (2 epochs) compared to TD's 11 days, both on a single A6000. Furthermore, when limited to the same equivariance, CoarsenConf-OT performs predominantly better (Appendix Tab. 6). As CoarsenConf (autoregressive, without the OT loss) results in a lower overall DRUGS RMSD distribution (Fig. 4), we use this model for the remaining downstream tasks.

## 4.2 GEOM BENCHMARKS: PROPERTY PREDICTION

**Problem setup.** We generate and relax min(2L, 32) conformers (L ground truth) for 100 molecules from GEOM-DRUGS using GFN2-xTB with the BFGS optimizer. We then predict various properties, including Energy ($E$), HOMO-LUMO Gap ($\Delta\epsilon$), minimum Energy ($E_{min}$) in kcal/mol, and dipole moment ($\mu$) in debye via xTB (Bannwarth et al., 2019). The mean absolute error of the generated ensemble properties compared to ground truth is reported.

**Results.** Tab. 2 demonstrates CoarsenConf's ability to generate the lowest energy structures with the most accurate chemical properties. For further discussions, please see Appendix §H.

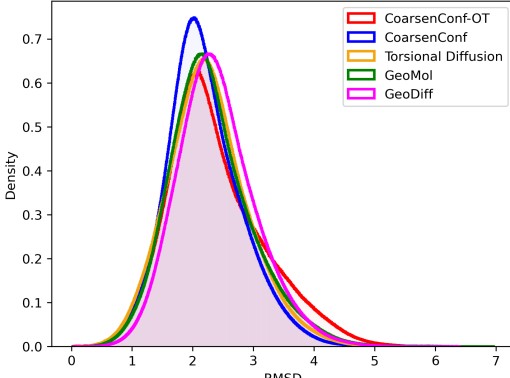

Table 2: Property prediction: Mean absolute error of generated vs. ground truth ensemble properties for $E$, HOMO-LUMO gap $\Delta\epsilon$, $E_{min}$ (kcal/mol), and dipole moment $\mu$ (debye) calculated with xTB.

| | $E$ | $\mu$ | $\Delta\epsilon$ | $E_{min}$ |
|---|---|---|---|---|
| DMCG | - | - | - | 0.136 |
| GeoDiff | - | - | - | 0.155 |
| GeoMol | 28.80 | 1.475 | 4.186 | 0.267 |
| Torsional Diffusion | 16.75 | 1.333 | 2.908 | 0.096 |
| CoarsenConf | **12.41** | **1.250** | **2.522** | **0.049** |

Figure 4: GEOM-DRUGS test set RMSD error distributions for each ML model.

## 4.3 FLEXIBLE ORACLE-BASED PROTEIN DOCKING

We evaluate MCG models, pretrained on GEOM-DRUGS, using nine protein docking oracle functions provided by the Therapeutics Data Commons (TDC) (Huang et al., 2021).

**Problem setup.** Starting with a known 2D ligand[2] molecule, protein, and desired 3D protein binding pocket, we measure conformer quality by comparing the predicted binding affinity of generated conformers of each MCG method. TDC's protein docking oracle functions take in a ligand SMILES string, generate a 3D conformer, and try multiple poses, before ultimately returning the best binding affinity via Autodock Vina's flexible docking simulation. We augment TDC with the ability to query ML models pretrained on GEOM-DRUGS, instead of the built-in RDKit + MMFF approach for MCG. For each evaluated MCG method, we generate 50 conformers for each of the nine ligands and report the best (lowest) binding affinity. Given that the ligand and protein identity and the protein pocket are fixed, this task measures the quality of 3D conformer coordinates through their binding efficacy to the specified pocket. We note that this task is indicative of real-world simulation workflows.

**Results.** CoarsenConf significantly outperforms prior MCG methods on the TDC oracle-based affinity prediction task ( Tab. 3). CoarsenConf generates the best ligand conformers for 8/9 tested proteins, with improvements of up to 53% compared to the next best method. CoarsenConf is 1.46 kcal/mol better than all methods when averaged over all 9 proteins, which corresponds to a 14.4% improvement on average compared to the next best method.

## 4.4 RIGID ORACLE-BASED PROTEIN DOCKING

We evaluate MCG models on rigid oracle-based protein docking. We use the 166000 protein-ligand complexes from the CrossDocked (Francoeur et al., 2020) training set (Appendix §K for more details).

**Problem setup.** Similar to the flexible docking task (§4.3), we can generate conformers of known ligands for known protein pockets, but now have them only undergo a rigid pocket-specific energy minimization before predicting the binding affinity. To handle the fact that MCG generates ligand structures in a vacuum and has no information about the target protein, we assume the centroid of the ground truth ligand is accurate, and translate each generated structure to match the center.

---

[2]No ground truth 3D structures. All ligand SMILES taken from Protein Data Bank: https://www.rcsb.org/

Table 3: Quality of best generated conformer for known protein ligands for all 9 proteins from the TDC library. Quality is measured by free energy change (kcal/mol) of the binding process with AutoDock Vina's flexible docking simulation ($\downarrow$ is better).

| | Best Protein-Conformer Binding Affinity ($\downarrow$ is better) | | | | | | | | |
|---|---|---|---|---|---|---|---|---|---|
| Method | 3PBL | 2RGP | 1IEP | 3EML | 3NY8 | 4RLU | 4UNN | 5M04 | 7L11 |
| RDKit + MMFF | -8.26 | -11.42 | -10.75 | -9.26 | -9.69 | -8.72 | -9.73 | -9.53 | -9.19 |
| GeoMol | -8.23 | -11.49 | -11.16 | -9.39 | **-11.66** | -8.85 | -10.28 | -9.31 | -9.29 |
| Torsional Diffusion | -8.53 | -11.34 | -10.76 | -9.25 | -10.32 | -8.96 | -10.65 | -9.61 | -9.10 |
| CoarsenConf | **-8.81** | **-12.93** | **-16.43** | **-9.82** | -11.26 | **-9.54** | **-11.62** | **-14.00** | **-9.43** |

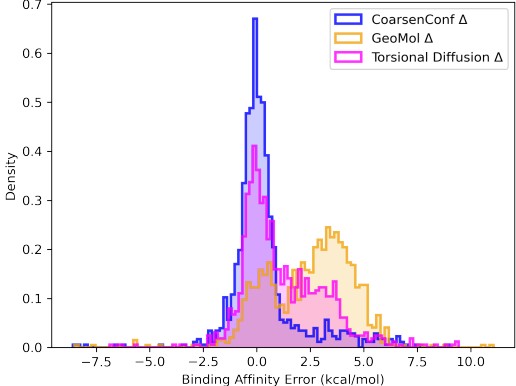

Table 4: Binding affinity error distribution statistics in kcal/mol (more negative is better).

| Method | Mean | Min |
|---|---|---|
| GeoMol | 2.476 | -8.523 |
| Torsional Diffusion | 1.178 | -6.876 |
| CoarsenConf | **0.368** | **-8.602** |

Figure 5: Binding affinity ($\downarrow$ is better) error distributions for 100k conformer-protein complexes in the CrossDocked dataset. The error is the difference in binding affinity between the generated and ground truth energy minimized 3D ligands.

We use AutoDock Vina (Eberhardt et al., 2021) and its local BFGS energy optimization (similar to that done with xTB), *i.e.* a relaxation of all tested structures (including the ground truth), following the SBDD benchmarking framework Guan et al. (2023). We report the difference between the generated structure's minimum binding affinity and that of the ground truth 3D ligand. Unlike the docking simulation, Vina's energy minimization does not directly adjust the torsion angles or internal degrees of freedom within the ligand. Instead, it explores different atomic positions and orientations of the entire ligand molecule within the binding site of the protein to find energetically favorable binding poses. By further isolating the MCG-generated structures, this task better evaluates the generative capacity of MCG models. While the original SBDD task (Peng et al., 2022; Guan et al., 2023) reports the Vina minimization score as its main metric, it requires the 3D ligand to be generated from scratch. Here, we use the SBDD framework to isolate the generated ligand conformer as the only source of variability to evaluate the biological and structural quality of MCG models in an unbiased fashion.

**Results.** We report the results for 100,000 unique conformer-protein interactions; note that there is a large cost to run the binding affinity prediction (see Appendix §K for more details). We also emphasize that the presented evaluation is not to be confused with actual docking solutions, as a low-energy conformer is not always guaranteed to be the best binding pose. Instead, we employ an unbiased procedure to present empirical evidence for how CoarsenConf can generate input structures to Vina that significantly outperform prior MCG models in achieving the best binding affinities.

Fig. 5 further demonstrates CoarsenConf's superior performance on orders of magnitude more protein complexes than the prior flexible oracle task. CoarsenConf decreases the average error by 56% compared to TD, and is the only method not to exhibit bimodal behavior with error greater than zero. We hypothesize that the success of MCG methods in matching ground truth structures is influenced by the complexity of protein pockets. In simpler terms, open pockets better facilitate Vina's optimization, but the initial position generated by MCG remains crucial. Furthermore, if the initial MCG-generated structure did not matter, the distributions for each MCG model would be identical. We also note that the lowest energy conformer is not always the optimal ligand structure for binding, but in our experiments, it yields the best input for the Vina-based oracles. Overall, CoarsenConf best approximates the ground truth ligand conformers of CrossDocked and generates the best structures for Vina's rigid energy relaxation and binding affinity prediction.

## 5 CONCLUSION

We present CoarsenConf, a novel approach for robust molecular conformer generation that combines an SE(3)-equivariant hierarchical VAE with geometric coarse-graining techniques for accurate conformer generation. By utilizing easy-to-obtain approximate conformations, our model effectively learns the optimal distortion to generate low-energy conformers. CoarsenConf possesses unrestricted degrees of freedom, as it can adjust atomic coordinates, distances, and torsion angles freely. CoarsenConf's CG procedure can also be tailored to handle even larger systems, whereas prior methods are restricted to full FG or torsion angle space. Our experiments demonstrate the effectiveness of CoarsenConf compared to existing methods. Our study also extends recent 3D molecule-protein benchmarks to conformer generation, providing valuable insights into robust generation and downstream biologically relevant tasks.

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

# A  RELATED WORK

**Autoregressive Molecule Generation.**  Autoregressive models provide control over the generative process by enabling direct conditioning on prior information, allowing for a more precise and targeted generation of output. Autoregressive generation has shown success in 2D molecule tasks using SMILE-based methods, as seen in MolMIM (Reidenbach et al., 2023), as well as graph-based atom-wise and subgraph-level techniques, as shown in GraphAF (Shi et al., 2020) and HierVAE (Jin et al., 2020). Similarly, 3DLinker (Huang et al., 2022) and SQUID (Adams and Coley, 2023) showcase the usefulness of 3D autoregressive molecule generation and their ability to leverage conditional information in both atom-wise and subgraph-level settings for 3D linkage and shape-conditioned generative tasks respectively. We note that, unlike prior methods (Adams and Coley, 2023), CoarsenConf does not require a predefined fragment vocabulary. HERN (Jin et al., 2022b) further demonstrates the power of hierarchical equivariant autoregressive methods in the task of computational 3D antibody design. Similarly, Pocket2Mol (Peng et al., 2022) uses autoregressive sampling for structure-based drug design.

**Protein Docking and Structure-based Drug Design.**  Protein docking is a key downstream use case for generating optimal 3D molecule structures. Recent research has prominently explored two distinct directions within this field. The first is blind docking, where the goal is to locate the pocket and generate the optimal ligand to bind (Corso et al., 2022). The second is structure-based drug design (SBDD), where optimal 3D ligands are generated by conditioning on a specific protein pocket. Specifically, the SBDD task focuses on the ability to generate ligands that achieve a low AutoDock Vina score for the CrossDocked2020 (Francoeur et al., 2020) dataset. AutoDock Vina (Eberhardt et al., 2021) is a widely used molecular docking software that predicts the binding affinity of ligands (drug-like molecules) to target proteins. Autodock Vina takes in the 3D structures of the ligand, target protein, and binding pocket and considers various factors such as van der Waals interactions, electrostatic interactions, and hydrogen bonding between the ligand and target protein to predict the binding affinity. We demonstrate how SBDD can be adapted to construct comprehensive MCG benchmarks. In this framework, we evaluate the generative abilities of MCG models by measuring the binding affinities of generated comforters and comparing them to the provided ground truth ligand conformers for a wide array of protein-ligand complexes.

**SE(3)-Equivariance.**  Let $\mathcal{X}$ and $\mathcal{Y}$ be the input and output vector spaces, respectively, which possess a set of transformations $G\colon G \times \mathcal{X} \to \mathcal{X}$ and $G \times \mathcal{Y} \to \mathcal{Y}$. The function $\phi : \mathcal{X} \to \mathcal{Y}$ is called equivariant with respect to $G$ if, when we apply any transformation to the input, the output also changes via the same transformation or under a certain predictable behavior, *i.e.*,

**Definition 1** *The function $\phi : \mathcal{X} \mapsto \mathcal{Y}$ is G-equivariant if it commutes with any transformation in $G$,*

$$\phi(\rho_{\mathcal{X}}(g)x) = \rho_{\mathcal{Y}}(g)\phi(x), \forall g \in G, \tag{4}$$

*where $\rho_{\mathcal{X}}$ and $\rho_{\mathcal{Y}}$ are the group representations in the input and output space, respectively. Specifically, $\phi$ is called invariant if $\rho_{\mathcal{Y}}$ is the identity.*

By enforcing SE(3)-equivariance in our probabilistic model, $p(\boldsymbol{X}|\mathcal{R})$ remains unchanged for any rototranslation of the approximate conformer $\mathcal{R}$. CoarsenConf's architecture is inspired by recent equivariant graph neural network architectures, such as EGNN (Satorras et al., 2021) and PaiNN (Schütt et al., 2021), as well as Vector Neuron multi-layer perceptron (VN-MLP) (Deng et al., 2021).

**Coarse-to-Fine.**  Coarse-to-fine generation has been recently explored in Qiang et al. (2023), where they leverage latent diffusion in coarse-grain space to generate new molecules, before decoding back to fine-grain space. Qiang et al. (2023) accomplishes state-of-the-art performance in generating molecules from scratch via a pre-defined fragment vocabulary, or a collection of the CG components of the molecule space they want to model (detailed in their Appendix A). This is a common approach for many fragment-based generation methods for 2D and 3D molecule generation. We note that CoarsenConf does not use such a vocabulary, and pulls all of its fragments directly from the input molecule based on the specified coarsening criteria, which can be updated easily. This means that if trying to generate an unseen fragment at inference time, vocabulary-based methods would fail and have to be retrained, whereas CoarsenConf can parse all inputs.

# B  LOSS FUNCTION

As described in §3, CoarsenConf optimizes the following loss function:

$$\text{MSE}(\mathcal{A}(X, X_{true})) + \beta_1 D_{KL}(q_\phi(z|X, \mathcal{R}) \parallel p_\psi(z|\mathcal{R})) + \beta_2 \frac{1}{|\mathcal{E}^*|} \sum_{(i,j) \in \mathcal{E}^*} ||r_{ij} - r_{ij}^{true}||^2, \quad (5)$$

where $\mathcal{A}$ is the Kabsch alignment function (Kabsch, 1993), $\mathcal{E}^*$ are all the 1 and 2-hop edges in the molecular graph, with $r_{ij}$ corresponding to the distance between atoms $i$ and $j$. We note that both $\beta_1$ and $\beta_2$ play a crucial role in the optimization. $\beta_1$ has to be set low enough ($1e{-}3$) to allow the optimization to focus on the MSE when the differences between the model-based $X$ and the ground truth are very close, due to the RDKit distortion parameterization.

For the QM9 experiments, $\beta_1$ is annealed starting from $1e{-}6$ to $1e{-}1$, increasing by a factor of 10 each epoch. $\beta_2$ controls the distance auxiliary loss and also had to be similarly annealed. We found that when $\beta_2 = 0$, CoarsenConf still learned to improve upon the aligned MSE loss by 50%, as compared to RDKit. Our error analysis showed that the resulting molecules either had extremely low distance error with high MSE, or vice-versa. Therefore, when the learning objective is unconstrained, our model learns to violate distance constraints by placing atoms in low-error but unphysical positions.

For QM9, by slowly annealing the distance loss, we allow our model to reach a metaphysical unstable transition state where distances are violated, but the aligned coordinate error is better. We then force the model to respect distance constraints. In the case of DRUGS, we found that this transition state was too difficult for the model to escape from, and we report the results using $\beta_2 = 0.5$ in Tab. 1. In Appendix §J, we further explore this idea and experiment with different annealing schedules for DRUGS. We note that as CoarsenConf learns the torsion angles in an unsupervised manner because of the chosen CG strategy, we leave explicit angle optimization to future work.

# C  COARSE-GRAINING

We elaborate on the coarse-graining procedure introduced in §3. Following Wang et al. (2022), we represent fine-grained (FG) molecular conformers as $x = \{x_i\}_{i=1}^n \in \mathbb{R}^{n \times 3}$. Similarly, the coarse-grained (CG) conformers are represented by $X = \{X_I\}_{I=1}^N \in \mathbb{R}^{N \times 3}$ where $N < n$. Let $[n]$ and $[N]$ denote the set $\{1, 2, ..., n\}$ and $\{1, 2, ..., N\}$ respectively. The CG operation can be defined as an assignment $m : [n] \to [N]$, which maps each FG atom $i$ in $[n]$ to CG bead $I \in [N]$, i.e., bead $I$ is composed of the set of atoms $C_I = (k \in n \mid m(k) = I)$. $X_I$ is initialized at the center of mass = $\frac{1}{|C_I|} \sum_{j \in C_I} x_j$.

We note that CoarsenConf coarsens input molecules by first severing all torsion angles $\tau_{abcd}$, with $k$ torsion angles resulting in $k + 1$ connected components or CG beads. This allows us, on average, to represent QM9 molecules with three beads and large drug molecules ($n > 100$) with 29 beads. We opted for a torsion angle-based strategy as it allows for unsupervised control over torsion angles, as well as the ability to rotate each subgraph independently. The CG strategy can be altered for various applications going forward.

# D  ENCODER EQUATIONS

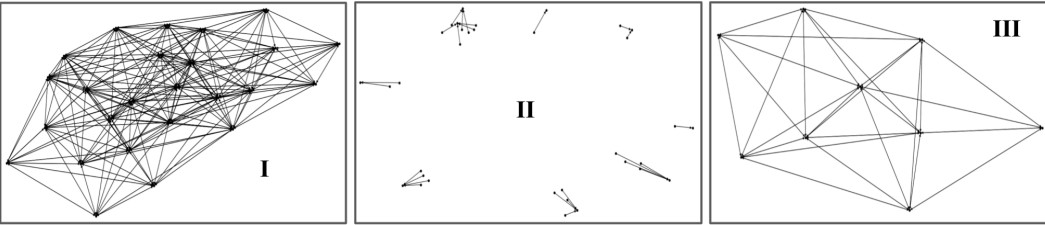

Figure 6: **Encoder module message passing structure. (I)** Fine-grained graph with auxiliary 4Å distance cut off. **(II)** Pooling graph with nodes for each atom and coarse-grained bead. Each group of nodes represents the formation of a CG bead. There is a single directional edge from each atom to its corresponding bead. **(III)** Coarse-grained graph with auxiliary 4Å distance cut off, using the learned representation from the pooling graph. CoarsenConf reduces the input from (I) to (III), drastically reducing the complexity of the problem.

## D.1  FINE-GRAIN MODULE

We describe the encoder, shown in Fig. 2(I). The model operates over SE(3)-invariant atom features $h \in R^{n \times D}$, and SE(3)-equivariant atomistic coordinates $x \in R^{n \times 3}$. A single encoder layer is composed of three modules: fine-grained, pooling, and coarse-grained. Full equations for each module can be found in Appendix §D.1, §D.2, §D.3, respectively.

The fine-grained module is a graph-matching message-passing architecture. It differs from Stärk et al. (2022) by not having internal closed-form distance regularization and exclusively using unidirectional attention. It aims to effectively match the approximate conformer and ground truth by updating attention from the former to the latter.

The FG module is responsible for processing the FG atom coordinates and invariant features. More formally, the FG is defined as follows:

$$
\begin{aligned}
\boldsymbol{m}_{j \to i} &= \phi^e(\boldsymbol{h}_i^{(t)}, \boldsymbol{h}_j^{(t)}, \|\boldsymbol{x}_i^{(t)} - \boldsymbol{x}_j^{(t)}\|^2, \boldsymbol{f}_{j \to i}), \forall (I, J) \in \mathcal{E} \cup \mathcal{E}', \\
\boldsymbol{u}_{j' \to i} &= a_{j' \to i} \boldsymbol{W} \boldsymbol{h}_{j'}^{(t)}, \forall i \in \mathcal{V}, j' \in \mathcal{V}', \\
\boldsymbol{m}_i &= \frac{1}{|\mathcal{N}(i)|} \sum_{j \in \mathcal{N}(i)} \boldsymbol{m}_{j \to i}, \forall i \in \mathcal{V} \cup \mathcal{V}', \\
\boldsymbol{u}_i &= \sum_{j' \in \mathcal{V}'} \boldsymbol{u}_{j' \to i}, \forall i \in \mathcal{V}, \quad \text{and} \quad \boldsymbol{u}_i' = 0, \\
\boldsymbol{x}_i^{(t+1)} &= \eta_x \cdot \boldsymbol{x}_i^{(0)} + (1 - \eta_x) \cdot \boldsymbol{x}_i^{(t)} + \sum_{j \in \mathcal{N}(i)} (\boldsymbol{x}_i^{(t)} - \boldsymbol{x}_j^{(t)}) \phi^x(\boldsymbol{m}_{j \to i}), \\
\boldsymbol{h}_i^{(t+1)} &= (1 - \eta_h) \cdot \boldsymbol{h}_i^{(t)} + \eta_h \cdot \phi^h(\boldsymbol{h}_i^{(t)}, \boldsymbol{m}_i, \boldsymbol{u}_i, \boldsymbol{f}_i), \forall i \in \mathcal{V} \cup \mathcal{V}',
\end{aligned} \tag{6}
$$

where $f$ represents the original invariant node features $h^{t=0}$, $a_{j \to i}$ are SE(3)-invariant attention coefficients derived from $h$ embeddings, $\mathcal{N}(i)$ are the graph neighbors of node $i$, and $W$ is a parameter matrix. $(\mathcal{V}, \mathcal{E})$ and $(\mathcal{V}', \mathcal{E}')$ refer to the low-energy and RDKit approximation molecular graphs, respectively. The various $\phi$ functions are modeled using shallow MLPs, with $\phi^x$ outputting a scalar and $\phi^e$ and $\phi^h$ returning a D-dimensional vector. $\eta_x$ and $\eta_h$ are weighted update parameters for the FG coordinates $x$ and invariant features $h$ respectively. We note that attention flows in a single direction from the RDKit approximation to the ground truth to prevent leakage in the parameterization of the learned prior distribution.

## D.2  POOLING MODULE

The pooling module takes in the updated representations ($h$ and $x$) of both the ground truth molecule and the RDKit reference from the FG module. The pooling module is similar to the FG module, except it no longer uses attention and operates over a pooling graph. Given a molecule with $n$ atoms

and $N$ CG beads, the pooling graph consists of $n + N$ nodes. There is a single directional edge from all atoms to their respective beads. This allows message passing to propagate information through the predefined coarsening strategy.

The pooling module is responsible for learning the coordinates and invariant features of each coarse-grained bead by pooling FG information in a graph-matching framework. More formally, the pooling module is defined as follows:

$$
\begin{aligned}
\boldsymbol{m}_{j \to I} &= \phi^e(\boldsymbol{H}_I^{(t)}, \boldsymbol{h}_j^{(t)}, \|\boldsymbol{X}_I^{(t)} - \boldsymbol{x}_j^{(t)}\|^2, \boldsymbol{f}_{j \to I}), \forall (I, J) \in \mathcal{E} \cup \mathcal{E}', \\
\boldsymbol{m}_I &= \frac{1}{|\mathcal{N}(I)|} \sum_{j \in \mathcal{N}(I)} \boldsymbol{m}_{j \to I}, \forall I \in \mathcal{V} \cup \mathcal{V}', \\
\boldsymbol{X}_I^{(t+1)} &= \eta_X \cdot \boldsymbol{X}_I^{(0)} + (1 - \eta_X) \cdot \boldsymbol{X}_I^{(t)} + \sum_{j \in \mathcal{N}(I)} (\boldsymbol{X}_I^{(t)} - \boldsymbol{x}_j^{(t)}) \phi^x(\boldsymbol{m}_{j \to I}), \\
\boldsymbol{H}_I^{(t+1)} &= (1 - \eta_H) \cdot \boldsymbol{H}_I^{(t)} + \eta_H \cdot \phi^h(\boldsymbol{H}_I^{(t)}, \boldsymbol{m}_I, \boldsymbol{f}_I), \forall I \in \mathcal{V} \cup \mathcal{V}',
\end{aligned}
\tag{7}
$$

where capital letters refer to the CG representation of the pooling graph. The pooling module mimics the FG module without attention on a pooling graph, as seen in Fig. 6(II). The pooling graph contains a single node for each atom and CG bead, with a single edge from each FG atom to its corresponding bead. It is used to learn the appropriate representations of the CG information. As the pooling graph only contains edges from fine-to-coarse nodes, the fine-grain coordinates and features remain unchanged. The pooling graph at layer $t$ uses the invariant feature $H$ from the CG module of layer $t - 1$ to propagate information forward through the neural network. The main function of the pooling module is to act as a buffer between the FG and CG spaces. As a result, we found integrating the updated CG representation useful for building a better transition from FG to CG space.

### D.3   COARSE-GRAIN MODULE

The coarse-grained module uses the updated CG representations ($H \in R^{N \times D}$ and $X \in R^{N \times 3}$) from the pooling module to learn equivariant CG features ($Z$ and $\tilde{Z} \in R^{N \times F \times 3}$) for the ground truth molecule and the RDKit reference. $F$ is fixed as a hyperparameter for latent space size. $N$ is allowed to be variable-length to handle molecules resulting from any coarsening procedure. The CG features are learned using a graph-matching point convolution (Thomas et al., 2018) with similar unidirectional attention as the FG module. Prior to the main message-passing operations, the input features undergo equivariant mixing (Huang et al., 2022) to further distill geometric information into the learned CG representation.

The CG module is responsible for taking the pooled CG representation from the pooling module and learning a node-level equivariant latent representation. We note that we use simple scalar and vector operations to mix equivariant and invariant features without relying on computationally expensive higher-order tensor products. In the first step, invariant CG features $H$ and equivariant features $\boldsymbol{v} \in \mathbb{R}^{F \times 3}$ are transformed and mixed to construct new expressive intermediate features $H', H'', \boldsymbol{v}'$ by,

$$
H'_I = \phi_1(h_I^{(t)}, \|\text{VN-MLP}_1(\boldsymbol{v}_I^{(t)})\|) \in \mathbb{R}^D, \tag{8a}
$$

$$
H''_I = \phi_2(h_I^{(t)}, \|\text{VN-MLP}_2(\boldsymbol{v}_I^{(t)})\|) \in \mathbb{R}^F, \tag{8b}
$$

$$
\boldsymbol{v}'_I = \text{diag}\{\phi_3(H_I^{(t)})\} \cdot \text{VN-MLP}_3(\boldsymbol{v}_I^{(t)}) \in \mathbb{R}^{F \times 3}. \tag{8c}
$$

Next, a point convolution (Thomas et al., 2018; Schütt et al., 2021; Huang et al., 2022) is applied to linearly transform the mixed features $H'$, $H''$, $\boldsymbol{v}'$ into messages:

$$\boldsymbol{m}^{\boldsymbol{H}}_{I \leftarrow J} = \text{Ker}_1(\|\boldsymbol{r}_{I,J}\|) \odot H'_J, \tag{9a}$$

$$\boldsymbol{m}^{\boldsymbol{v}}_{I \leftarrow J} = \text{diag}\left\{\text{Ker}_2(\|\boldsymbol{r}_{I,J}\|)\right\} \cdot \boldsymbol{v}'_J + \left(\text{Ker}_3(\|\boldsymbol{r}_{I,J}\|) \odot H''_J\right) \cdot \boldsymbol{r}^{\top}_{I,J}, \tag{9b}$$

$$\boldsymbol{u}_{J' \to I} = a_{J' \to I} \boldsymbol{W} \boldsymbol{H}^{(t)}_{J'}, \forall I \in \mathcal{V}, J' \in \mathcal{V}', \tag{9c}$$

$$\boldsymbol{u}_I = \sum_{J' \in \mathcal{V}'} \boldsymbol{u}_{J' \to I}, \forall I \in \mathcal{V}, \quad \text{and} \quad \boldsymbol{u}'_I = 0, \tag{9d}$$

$$\boldsymbol{H}^{t+1}_I = (1 - \eta_H) \cdot H^{\ell}_I + \eta_H \cdot \text{MLP}(H^{\ell}_I, \sum_{J \in N(I)} m^H_{I \leftarrow J}, u_I), \forall I \in \mathcal{V} \cup \mathcal{V}', \tag{9e}$$

$$\boldsymbol{v}^{t+1}_I = (1 - \eta_v) \cdot v^{\ell}_I + \eta_v \cdot \text{VN-MLP}_4(\boldsymbol{v}^{\ell}_I, \sum_{J \in N(I)} \boldsymbol{m}^{\boldsymbol{v}}_{I \leftarrow J}), \forall I \in \mathcal{V} \cup \mathcal{V}', \tag{9f}$$

where each Ker refers to a learned RBF kernel, $r_{IJ}$ is the difference between $X_I$ and $X_J$, and $a_{J \to I}$ are SE(3)-invariant attention coefficients derived from the learned invariant features $H$. $\eta_H$ and $\eta_v$ control the mixing of the learned invariant and equivariant representations.

We note that for $t > 0$, the $H_I$ from the CG module are used in the next layer's pooling module, creating a cyclic dependency to learn an information-rich CG representation. This is shown by the dashed lines in Fig. 2(I). The cyclic flow of information grounds the learned CG representation to the innate FG structure. All equivariant CG features $v$ are initialized as zero and are slowly built up through each message passing layer. As point convolutions and VN operations are strictly SO(3)-equivariant, we subtract the molecule's centroid from the atomic coordinates prior to encoding, making it effectively SE(3)-equivariant.

The modules in each encoder layer communicate with the respective module of the previous layer. This hierarchical message-passing scheme results in an informative and geometrically grounded final CG latent representation. We note that the pooling module of layer $\ell$ uses the updated invariant features $H$ from the CG module of layer $\ell - 1$, as shown by the dashed lines in Fig. 2(I).

## D.4 ALGORITHMS

We include algorithms that define the forward pass of the encoder, as well as the training and inference procedure.

---

**Algorithm 1** Encoder Forward Pass: Hierarchical Message Passing Inputs and Outputs

---

1: true coord $X$, RDKit coord $\hat{X}$
   pooing coord $X_p$, pooling RDKit coord $\hat{X}_p$
   coarse coord $X_c$, coarse RDKit coord $\hat{X}_c$
   true features $h$, RDKit features $\hat{h}$
   pooing features $h_p$, pooling RDKit features $\hat{h}_p$
   coarse features $h_c$, coarse RDKit features $\hat{h}_c$
   true latent CG representation $Z$, RDKit latent CG representation $\hat{Z}$
2: $(FG), (PL), (CG) \leftarrow$ dataloader$[i]$ // *Fine-grain, Pooling, and Coarse-grain graphs*
3: $(X, h), (\hat{X}, \hat{h}) \leftarrow FG$ // *$X \in \mathbb{R}^{n \times 3}$ and $h \in \mathbb{R}^{n \times D}$*
4: $(X_p, h_p), (\hat{X}_p, \hat{h}_p) \leftarrow PL$ // *$X_p \in \mathbb{R}^{n+N \times 3}$ and $h_p \in \mathbb{R}^{n+N \times D}$*
5: $(X_c, h_c), (\hat{X}_c, \hat{h}_c) \leftarrow CG$ // *$X_c \in \mathbb{R}^{N \times 3}$ and $h_c \in \mathbb{R}^{N \times D}$*
6: $Z, \hat{Z} \leftarrow 0, 0$ // *$Z = [v_I \; \forall I \in X_c]$, init as zeros $\in \mathbb{R}^{N \times F \times 3}$, see Eq. 8*
7: **for** $t$ in num_layers **do**
8:    $(X, h), (\hat{X}, \hat{h}) \leftarrow$ FG_Module$((X, h), (\hat{X}, \hat{h}))$ // *see Eq. 6*
9:    $X_p[0:n] \leftarrow X$
10:    $\hat{X}_p[0:n] \leftarrow \hat{X}$
11:    $h_p[0:n] \leftarrow h$
12:    $\hat{h}_p[0:n] \leftarrow \hat{h}$ // *Set pooling graphs features with output of FG Module*
13:    $X_p[n:n+N] \leftarrow X_c$
14:    $\hat{X}_p[n:n+N] \leftarrow \hat{X}_c$
15:    $h_p[n:n+N] \leftarrow h_c$
16:    $\hat{h}_p[n:n+N] \leftarrow \hat{h}_c$ // *Set pooling graphs features with output of CG Module*
17:    $(X_p, h_p), (\hat{X}_p, \hat{h}_p) \leftarrow$ Pooling_Module$((X_p, h_p), (\hat{X}_p, \hat{h}_p))$ // *see Eq. 7*
18:    $X_c \leftarrow X_p[n:n+N]$
19:    $\hat{X}_c \leftarrow \hat{X}_p[n:n+N]$
20:    $h_c \leftarrow h_p[n:n+N]$
21:    $\hat{h}_c \leftarrow \hat{h}_p[n:n+N]$ // *Set CG graphs features with output of Pooling Module*
22:    $Z, \hat{Z} \leftarrow$ CG_Module$((X_c, h_c), (\hat{X}_c, \hat{h}_c), Z, \hat{Z})$ // *[see Eq. 8 & Eq. 9]*
23: **end for**
24: **Return** $Z, \hat{Z} \in \mathbb{R}^{N \times F \times 3}$

---

We draw attention to the setting of pooling and coarse features before each of the respective modules (Lines 8-16 and 18-21). This is what sets the inputs to each layer and defines the outputs.

---

**Algorithm 2** Training One Epoch

---

1: **for** data in dataloader **do**
2:    $(FG), (PL), (CG) \leftarrow$ data
3:    $(X, h), (\hat{X}, \hat{h}) \leftarrow FG$
4:    $Z, \hat{Z} \leftarrow$ Encoder$(data)$ // *Refer to Algorithm 1*
5:    $Z \leftarrow$ Posterior$(Z, \hat{Z})$ // *see Eq. 2*
6:    $\hat{Z} \leftarrow$ Prior$(\hat{Z})$ // *see Eq. 2*
7:    $X \leftarrow$ Aggregated_Attention$(\text{KV} = Z, \text{Q} = \hat{X})$
8:    $X_{Gen} \leftarrow$ Decoder$(X)$
9:    $loss \leftarrow$ Loss$(X, X_{\text{Gen}}) + \text{KL}(Z, \hat{Z})$ // *see Eq. 5*
10:    $loss$.backward$()$
11: **end for**

---

---

**Algorithm 3** Inference

---

1: **for** data in dataloader **do**
2:     $(FG), (PL), (CG) \leftarrow$ data
3:     $(X, h), (\hat{X}, \hat{h}) \leftarrow FG$
4:     $FG \leftarrow ((\hat{X}, \hat{h}), (\hat{X}, \hat{h}))$ *// use RDKit for both, discard true structure in inference*
5:     $PL \leftarrow (PL[1], PL[1])$ *// discard true structure in inference*
6:     $CG \leftarrow (CG[1], CG[1])$ *// discard true structure in inference*
7:     None, $\hat{Z} \leftarrow$ Encoder$((FG, PL, CG))$ *// Refer to Algorithm 1*
8:     $Z \leftarrow$ Prior$(\hat{Z})$ *// see Eq. 2*
9:     $X \leftarrow$ Aggregated_Attention$(KV = Z, Q = \hat{X})$
10:     $X_{Gen} \leftarrow$ Decoder$(X)$
11:     results $\leftarrow X_{Gen}$
12: **end for**
13: **return** results

---

## E   DECODER ARCHITECTURE

We sample from the learned posterior (training) and learned prior (inference) to get $Z = \mu + \epsilon\sigma$, where $\epsilon$ is noise sampled from a standard Gaussian distribution as the input to the decoder. We note the role of the decoder is two-fold. The first is to convert the latent coarsened representation back into FG space through a process we call channel selection. The second is to refine the fine-grain representation autoregressively to generate the final low-energy coordinates.

**Channel Selection.**   To explicitly handle all choices of coarse-graining techniques, our model performs variable-length backmapping. This aspect is crucial because every molecule can be coarsened into a different number of beads, and there is no explicit limit to the number of atoms a single bead can represent. Unlike CGVAE (Wang et al., 2022), which requires training a separate model for each choice in granularity $N$, CoarsenConf is capable of reconstructing FG coordinates from any $N$ (illustrated in Fig. 2(III)).

CGVAE defines the process of channel selection as selecting the top $k$ latent channels, where $k$ is the number of atoms in a CG bead of interest. Instead of discarding all learned information in the remaining $F - k$ channels in the latent representation, we use a novel aggregated attention mechanism. This mechanism learns the optimal mixing of channels to reconstruct the FG coordinates and is illustrated in Fig. 3. The attention operation allows us to actively query our latent representation for the number of atoms we need, and draw upon similarities to the learned RDKit approximation that has been distilled into the latent space through the encoding process. Channel selection translates the CG latent tensor $Z \in R^{N \times F \times 3}$ into FG coordinates $x_{cs} \in R^{n \times 3}$.

**Coordinate Refinement.**   Once channel selection is complete, we have effectively translated the variable-length CG representation back into the desired FG form. From here, $x_{cs}$ is grouped into its corresponding CG beads but left in FG coordinates to do a bead-wise autoregressive generation of final low-energy coordinates (Fig. 2(IV)). As there is no intrinsic ordering of subgraphs, we use a breadth-first search that prioritizes larger subgraphs with large out-degrees. In other words, we generate a linear order that focuses on the largest, most connected subgraphs and works outward. We believe that by focusing on the most central component first, which occupies the most 3D volume, we can reduce the propagation of error that is typically observed in autoregressive approaches. We stress that by coarse-graining by torsion angle connectivity, our model learns the optimal torsion angles in an unsupervised manner, as the conditional input to the decoder is not aligned. CoarsenConf ensures each next generated subgraph is rotated properly to achieve a low coordinate and distance error.

**Learning the Optimal Distortion.**   The decoder architecture is similar to the EGNN-based FG layer in the encoder. However, it differs in two important ways. First, we mix the conditional coordinates with the invariant atom features using a similar procedure as in the CG layer instead of typical graph matching. Second, we learn to predict the difference between the RDKit reference and ground truth conformations. This provides an upper error bound and enables us to leverage easy-to-obtain approximations more effectively.

More formally, a single decoder layer is defined as follows:

$$\boldsymbol{\mu}^{(t)} = \frac{1}{|\mathcal{V}_{prev}|} \sum_{k \in \mathcal{V}_{prev}} x_k, \tag{10a}$$

$$\tilde{\boldsymbol{h}}_i = \phi^m(\boldsymbol{h}_i^{(t)}, \boldsymbol{x}_i^{(t)}, \boldsymbol{\mu}^{(t)}, \|\boldsymbol{x}_i^{(t)} - \boldsymbol{\mu}^{(t)}\|^2), \forall i \in \mathcal{V}_{cur}, \tag{10b}$$

$$\boldsymbol{m}_{j \rightarrow i} = \phi^e(\tilde{\boldsymbol{h}}_i^{(t)}, \tilde{\boldsymbol{h}}_j^{(t)}, \|\boldsymbol{x}_i^{(t)} - \boldsymbol{x}_j^{(t)}\|^2, \|\boldsymbol{x}_i^{(t)} - \boldsymbol{x}_{ref,j}^{(t)}\|^2, \|\boldsymbol{x}_i^{(t)} - \boldsymbol{x}_{ref,i}^{(t)}\|^2), \forall (i,j) \in \mathcal{E}_{cur}, \tag{10c}$$

$$\boldsymbol{m}_i = \frac{1}{|\mathcal{N}(i)|} \sum_{j \in \mathcal{N}(i)} \boldsymbol{m}_{j \rightarrow i}, \forall i \in \mathcal{V}_{cur}, \tag{10d}$$

$$\boldsymbol{u}_{j' \rightarrow i} = a_{j' \rightarrow i} \boldsymbol{W} \boldsymbol{h}_{j'}^{(t)}, \forall i \in \mathcal{V}_{cur}, j' \in \mathcal{V}_{prev}, \tag{10e}$$

$$\boldsymbol{u}_i = \sum_{j' \in \mathcal{V}_{prev}} \boldsymbol{u}_{j' \rightarrow i}, \forall i \in \mathcal{V}_{cur}, \tag{10f}$$

$$\boldsymbol{x}_i^{(t+1)} = \boldsymbol{x}_{ref,i}^{(t)} + \sum_{j \in \mathcal{N}(i)} (\boldsymbol{x}_i^{(t)} - \boldsymbol{x}_j^{(t)}) \phi^x(\boldsymbol{m}_{j \rightarrow i}), \forall i \in \mathcal{V}_{cur}, \tag{10g}$$

$$\boldsymbol{h}_i^{(t+1)} = (1 - \beta) \cdot \boldsymbol{h}_i^{(t)} + \beta \cdot \phi^h(\tilde{\boldsymbol{h}}_i^{(t)}, \boldsymbol{m}_i, \boldsymbol{u}_i, \boldsymbol{f}_i), \forall i \in \mathcal{V}_{cur}, \tag{10h}$$

where $(\mathcal{V}_{cur}, \mathcal{E}_{cur})$ and $(\mathcal{V}_{prev}, \mathcal{E}_{prev})$ refer to the subgraph currently being generated and the set of all previously generated subgraphs, *i.e.*, the current state of the molecule. $\phi^m, \phi^e, \phi^x$, and $\phi^h$ refer to separate shallow MLPs for the feature mixing, edge message calculation, coordinate update, and invariant feature update, respectively. Eq. 10(a-b) creates a mixed feature for each atom consisting of the current FG invariant feature and 3D position vectors ($h$ and $x$), and the previous centroid $\mu$ and respective centroid distances. Eq. 10(c-d) defines the message passing operation that uses the aforementioned mixed features $\tilde{h}$ and a series of important distances between the model-based conformer and RDKit reference.Eq. 10(e-f) apply the same unidirectional attention updates seen in the encoder architecture. Eq. 10(g-h) update the position and feature vector for each atom using the above messages and attention coefficients, with $f$ representing the original invariant node features $h^{\ell=0}$ and $\beta$ a weighted update parameter. We emphasize that Eq. 10(g) formulates the overall objective as learning the optimal distortion of the RDKit reference to achieve the low-energy position *i.e.*, $x^* = x_{ref} + \Delta x$. The CG autoregressive strategy allows CoarsenConf to handle extremely large molecules efficiently, as the max number of time steps is equal to the max number of CG beads. CoarsenConf is trained using teacher forcing (Williams and Zipser, 1989), which enables an explicit mixing of low-energy coordinates with the current FG positions from channel selection Eq. 10(a-b).

## F  MODEL CONFIGURATION

**Model.**  We present the model configuration that was used to generate the results in §4.1 - §4.4. Overall, the default model has 1.9M parameters: 1.6M for the encoder and 300K for the decoder. We note that as CoarsenConf uses graph matching, half the encoder parameters are used for each of the two inputs representing the same molecule in different spatial orientations. For both the encoder and decoder, we use five message-passing layers, a learning rate of $1e-3$ with an 80% step reduction after each epoch, and a latent space channel dimension ($F$) of 32. All other architectural parameters, such as feature mixing ratios or nonlinearities, were set following similar architectures (Huang et al., 2022; Deng et al., 2021; Stärk et al., 2022). We present further ablations in Appendix §J. We note that the ability to share weights between the inputs as well as between each layer in the encoder is left as a hyperparameter. This could allow the encoder to see a 2x or 5x reduction in model size, respectively.

**Compute.**  The QM9 model was trained and validated for five epochs in 15 hours using a single 40GB A100 GPU. We used a batch size of 600, where a single input refers to two graphs: the ground truth and RDKit approximate conformer. The DRUGs model was trained and validated for five epochs in 50 hours using distributed data-parallel (DDP) with 4 40GB A100 GPUs with a batch size of 300 on each GPU. For DRUGs, the GPU utilization was, on average, 66% as few batches contain very large molecules. In the future, lower run times can be achieved if the large molecules are more intuitively spaced out in each batch.

We note DDP has a negative effect on overall model benchmark performance due to the gradient synchronization but was used due to compute constraints. Without DDP, we expect the training time to take around 7 days, which is on par with Torsional Diffusion (4-11 days). We demonstrated that CoarsenConf achieves as good or better results than prior methods with less data and time, and these results can be further optimized in future work. We provide evidence of the negative effects of DDP in Appendix §J.

**Optimal Transport reduces compute requirements.** The optimal transport (OT) models were trained on 2 epochs on a single A6000 GPU for 8 and 15 hours total for QM9 and DRUGS, respectively. For OT details, see Appendix Eq. 11. Here, both models use the first 5 ground truth conformers. In real-world applications like polymer design, the availability of data is frequently limited and accompanied by a scarcity of conformers for each molecule. The current datasets, QM9 and DRUGS, do not mimic this setting very well. For example, on average, QM9 has 15 conformers per molecule, and DRUGS has 104 per molecule—both datasets have significantly more conformers than in an experimental drug design setting. Given this, rather than training on the first 30 conformers as done in Torsional Diffusion, we train on the first five (typically those with the largest Boltzmann weight) for QM9 and DRUGS, respectively.

## G RDKIT APPROXIMATE CONFORMERS

**Generating Approximate Conformers.** For CoarsenConf's initial conditional approximations, we only use RDKit + MMFF when it can converge (~90% and ~40% convergence for QM9 and DRUGS, respectively). We emphasize that RDKit only throws an error when MMFF is not possible but often returns structures with a non-zero return code, which signifies incomplete and potentially inaccurate optimizations. Therefore, in generating the RDKit structures for training and evaluation, we filter for MMFF converged structures. We default to the base EKTDG-produced structures when either the optimization cannot converge, or MMFF does not yield enough unique conformers. CoarsenConf ultimately offers a solution that can effectively learn from traditional cheminformatics methods. This aspect of MMFF convergence has not been discussed in prior ML for MCG methods, and we leave it to future cheminformatics research to learn the causes and implications of incomplete optimizations.

**Eliminating distribution shift with explicit conditioning.** Both CoarsenConf and TD optimize $p(X|\mathcal{R})$ but utilize the RDKit approximations $\mathcal{R}$ in different ways. TD learns to update the torsion angles of $\mathcal{R}$, while CoarsenConf leverages CG information to inform geometric updates (coordinates, distances, and torsion angles) to translate $\mathcal{R}$ to $X$. Unlike TD, which uses a preprocessing optimization procedure to generate substitute ground truth conformers that mimic $p(\mathcal{R})$, CoarsenConf directly learns from both $X$ and $\mathcal{R}$ through its hierarchical graph matching procedure. This directly addresses the distributional shift problem. We hypothesize that this, along with our angle-based CG strategy, leads to our observed improvements. Overall, CoarsenConf provides a comprehensive framework for accurate conformer generation that can be directly applied for downstream tasks such as oracle-based protein docking.

## H    GEOM BENCHMARK DISCUSSION

**xTB energy and property prediction.**    We note the issues surrounding the RMSD metrics have always existed, and prior MCG methods have introduced energy-based benchmarks that we describe and report in Tab. 2.These energies are calculated with xTB, and thus are not very accurate compared to density functional theory (DFT), as it is limited by the level of theory used to produce the energies further discussed in Axelrod and Gómez-Bombarelli (2022). Therefore, since current benchmarks mainly focus on gauging the effectiveness of the machine learning objective and less on the chemical feasibility and downstream use of the generated conformers, we use oracle-based protein docking-based to evaluate conformer quality on downstream tasks. These evaluations are highly informative, as molecular docking is a crucial step in the drug discovery process, as it helps researchers identify potential drug candidates and understand how they interact with their target proteins. The combination of RMSD, xTB energy, and downstream docking tasks presents a more comprehensive evaluation of generated conformers. GeoDiff and DMCG (Zhu et al., 2022) $E_{min}$ values were taken from their respective publications, as other properties are not directly comparable as measuring different properties with PSI4 (Smith et al., 2020) rather than xTB.

## I    QM9 EXPERIMENTAL DETAILS

Both CoarsenConf and CoarsenConf-OT were trained on 5 conformers per ground truth molecule, compared to Torsional Diffusion's 30. We hypothesize that since CoarsenConf uses a one-to-one loss function, we are able to maintain high recall, whereas the OT model finds an optimal matching that focuses on precision. By adding more ground truth conformers, we hypothesize our model can better cover the true conformer space, improving recall, as the OT setting would not be as biased toward precision.

Table 5: Quality of generated conformer ensembles for the GEOM-QM9 test set ($\delta = 0.5$Å) in terms of Coverage (%) and Average RMSD (Å). Torsional Diffusion (TD) was benchmarked using its evaluation code and available generated molecules, per their public instructions. Note that CoarsenConf (5 epochs) was restricted to using 41% of the data used by TD (250 epochs) to exemplify a low-compute and data-constrained setting. OMEGA results were taken from Jing et al. (2022) (we were unable to run the coverage normalization).

| | Recall | | | | Precision | | | |
| | Coverage ↑ | | AR ↓ | | Coverage ↑ | | AR ↓ | |
| Method | Mean | Med | Mean | Med | Mean | Med | Mean | Med |
|---|---|---|---|---|---|---|---|---|
| GraphDG* | 73.3 | 84.21 | 0.425 | 0.297 | 43.9 | 35.33 | 0.581 | 0.582 |
| CGCF* | 78.1 | 82.5 | 0.422 | 0.390 | 36.5 | 33.6 | 0.662 | 0.643 |
| ConfVAE* | 80.4 | 85.3 | 0.407 | 0.389 | 38.0 | 34.7 | 0.622 | 0.609 |
| DMCG* | 93.6 | 99.3 | 0.208 | 0.202 | 87.3 | 91.0 | 0.287 | 0.293 |
| OMEGA | 85.5 | 100.0 | 0.177 | 0.126 | 82.9 | 100.0 | 0.224 | 0.186 |
| RDKit + MMFF | 75.2 | 100.0 | 0.219 | 0.173 | 82.1 | 100.0 | 0.157 | 0.119 |
| GeoDiff | 76.5 | 100.0 | 0.297 | 0.229 | 50.0 | 33.5 | 0.524 | 0.510 |
| GeoMol | 79.4 | 100.0 | 0.219 | 0.191 | 75.9 | 100.0 | 0.262 | 0.233 |
| Torsional Diffusion | 82.2 | 100.0 | 0.179 | 0.148 | 78.4 | 100.0 | 0.222 | 0.197 |
| CoarsenConf | 76.9 | 100.0 | 0.246 | 0.211 | 80.2 | 100.0 | 0.227 | 0.186 |
| CoarsenConf-OT | 56.1 | 50.0 | 0.361 | 0.345 | 80.2 | 100.0 | 0.149 | 0.108 |

## J    DRUGS EXTENDED BENCHMARKS

**Evaluation Details.**    All models in Tab. 1 were benchmarked with Torsional Diffusion's (TD) evaluation code and retrained if generated molecules were not public (using their public instructions). We note that TD uses higher-order tensor products to maintain equivariance ($\ell = 2$). In contrast, GeoMol, GeoDiff, and CoarsenConf use scalar-vector operations that are theoretically analogous to $\ell = 1$. CoarsenConf-OT uses an optimal transport (OT) loss with the same decoder architecture as in Fig. 2, but is no longer autoregressive. GeoDiff's code would not load, so we were able to evaluate

the GeoDiff generated DRUGS molecules from the Torsional Diffusion authors' evaluation on the same test set.

Table 6: DRUGS-Precision equivariance ablations. OMEGA (Hawkins et al., 2010) results were taken from Jing et al. (2022). All others were re-benchmarked using Torsional Diffusion's code with an error normalized Coverage score to prevent the masking out of method failures. This enforces that each method is fairly evaluated on the entire test set as now Coverage truly represents the percentage of the test set that meets the threshold criteria. OMEGA requires a commercial license, so we were unable to test the results ourselves, thus taking results from TD. As a non-ML method, we also assume OMEGA has no failures as each molecule in the test set is valid, which could artificially inflate the observed coverage scores.

| Method | Coverage ↑ Mean | Coverage ↑ Med | AMR ↓ Mean | AMR ↓ Med |
|---|---|---|---|---|
| GraphDG* | - | - | 2.434 | 2.410 |
| CGCF* | - | - | 1.857 | 1.807 |
| ConfVAE* | - | - | 1.829 | 1.816 |
| ConfGF* | - | - | 1.730 | 1.711 |
| DMCG* | - | - | 0.921 | 0.879 |
| RDKit | 37.9 | 29.9 | 0.988 | 0.878 |
| RDKit + MMFF | 52.3 | 52.1 | 0.840 | 0.715 |
| OMEGA | 53.4 | 54.6 | 0.841 | 0.762 |
| GeoDiff | 23.7 | 13.0 | 1.131 | 1.083 |
| GeoMol | 40.5 | 33.5 | 0.919 | 0.842 |
| Torsional Diffusion ($\ell = 1$) | 48.9 | 50.0 | 0.804 | 0.758 |
| Torsional Diffusion ($\ell = 2$) | 52.1 | 53.7 | 0.770 | 0.720 |
| CoarsenConf | 43.8 | 35.5 | 0.914 | 0.829 |
| CoarsenConf-OT | 52.0 | 52.1 | 0.836 | 0.694 |

We copy the results from Tab. 1 and provide additional results, including TD for rotation order $\ell = 1$, OMEGA (Hawkins et al., 2010), and RDKit. This allows for a closer comparison to the scalar and vector operations that CoarsenConf employs to maintain equivariance. Using a lower rotation order results in slightly worse results in nearly all categories. We further discuss the implications of the choice in equivariant representation in Appendix §L.

The additional models with * denote that they were evaluated with a coverage threshold of 1.25Å instead of 0.75Å as done here. Our coverage threshold of 0.75Å is based on Jing et al. (2022). AMR is not a function of threshold and can be directly compared.

**Optimal Transport.** In practice, our model generates a set of conformers, $\{C_k\}_{k \in [1..K]}$, that needs to match a variable-length set of low-energy ground truth conformers, $\{C_l^*\}_{l \in [1..L]}$. In our case, the number L of true conformers, or the matching between generated and true conformers is not known upfront. For these reasons, we introduce an optimal transport-based, minimization-only, loss function (Ganea et al., 2021):

$$\mathcal{L}_{OT} = \min_{\mathbf{T} \in \mathcal{Q}_{K,L}} \sum_{k,l} T_{kl} \mathcal{L}(C_k, C_l^*),$$

(11)

$$\mathcal{L}(C_k, C_l^*) = \text{MSE}(C_k, C_l^*) + \text{distance error}(C_k, C_l^*),$$

where $\mathbf{T}$ is the ***transport plan*** satisfying $\mathcal{Q}_{K,L} = \{\mathbf{T} \in \mathbb{R}_+^{K \times L} : \mathbf{T}\mathbf{1}_L = \frac{1}{K}\mathbf{1}_K, \mathbf{T}^T\mathbf{1}_K = \frac{1}{L}\mathbf{1}_L\}$. The minimization w.r.t. T is computed quickly using the Earth Mover Distance and the POT library (Flamary et al., 2021). As the OT loss focuses more on finding the optimal mapping from generated conformers to ground truth reference, we removed the autoregressive decoding path of CoarsenConf and replaced it with a single pass with the same decoder architecture. The underlying loss function, which is tasked to minimize MSE coordinate error, and interatomic distance error is the same in both (Eq. 5), the autoregressive (AR) and non-AR OT-based loss functions. The OT version additionally finds the optimal mapping between the generated and ground truth structures, which better aligns with the AMR and Coverage benchmarks.

**Hyperparameter Ablations.** We experimented with increasing the latent channels ($F$) from 32 to 64 and 128, and introducing a step-wise distance loss and KL regularization annealing schedule, as done in the QM9 experiments. Both these experiments resulted in slightly worse performance when limited to 2 conformers per training molecule. We hypothesize that due to the DRUGs molecules being much larger than those in QM9, more training may be necessary, and a more sensitive annealing schedule may be required.

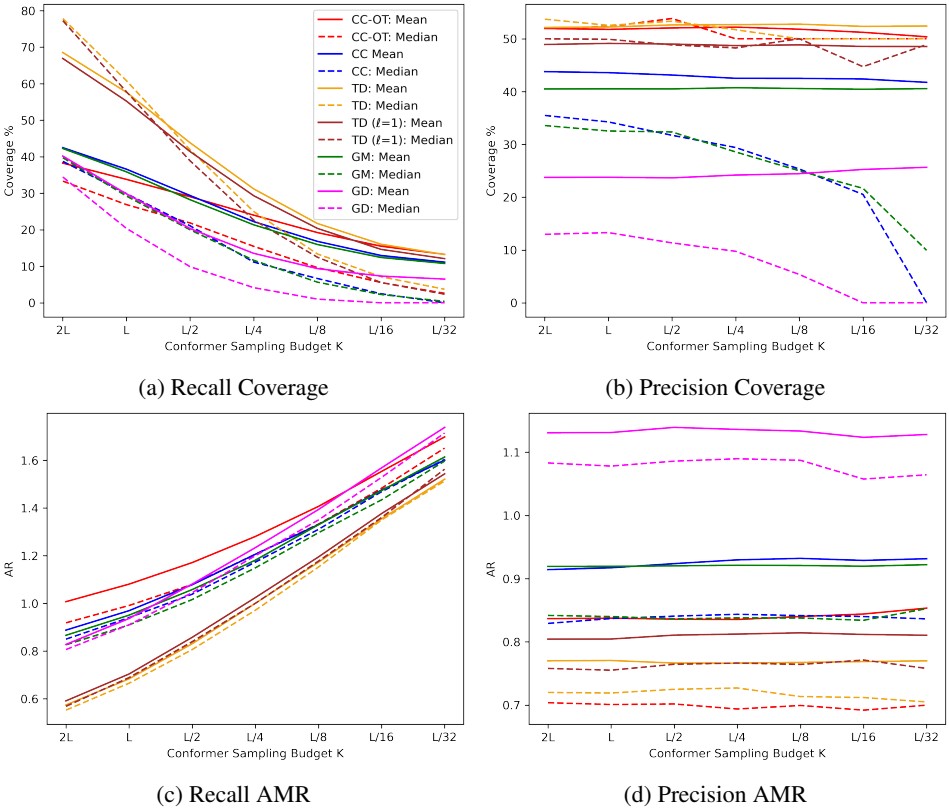

(a) Recall Coverage        (b) Precision Coverage

(c) Recall AMR        (d) Precision AMR

Figure 7: GEOM-DRUGS evaluation as a function of number of generated conformers. GEOM-DRUGS has 104 conformers per molecule on average. Recall is heavily dependent on the sampling budget. Precision is mostly stable. Lower AMR and higher coverage is better, but coverage is set by an arbitrary threshold, which in this case is 0.75Å. Results show CoarsenConf (CC), Torsional Diffusion (TD), GeoMol (GM), and GeoDiff (GD).

**GEOM-DRUGS Recall Results.** Fig. 7 demonstrates extensive Precision and Recall results for a wide range of tested sampling budgets for GEOM-DRUGS. We see that only Precision is stable across nearly all values. Due to the extreme sensitivity of the Recall metric and little difference in model performance for reasonable sampling budgets, we focus on Precision for QM9 and DRUGS. We also note that while CoarsenConf-OT saw worse recall results for QM9, this was not the case for DRUGS. In the case of DRUGS, CoarsenConf-OT achieves the learning objective of instilling force field optimizations as the lower error bound and does so with very little training and inference time.

# K    ORACLE-BASED PROTEIN DOCKING

We utilize the oracle-based protein docking task as molecules with higher affinity (more negative) have more potential for higher bioactivity, which is significant for real-world drug discovery. We use the CrossDocked2020 trainset consisting of 166000 protein-ligand interactions (2,358 unique proteins and 11,735 unique ligands) and its associated benchmarks, as it has been heavily used in Structure-based drug discovery as defined by Peng et al. (2022); Guan et al. (2023).

The CrossDocked2020 dataset is derived from PDBBind but uses smina (Koes et al., 2013), a derivative of AutoDock Vina with more explicit scoring control, to generate the protein-conditioned ligand structures to yield ground truth data. We note that based on the raw data, 2.2 billion conformer-protein interactions are possible, but we filtered out any ground-truth example that AutoDock Vina failed to score. Furthermore, in the TDC oracle-based task, each ligand is known to fit well in the given protein. CrossDocked2020, on the other hand, consists of various ligand-protein interactions, not all of which are optimal, making the overall task more difficult.

We note that while it takes on the orders of hours to generate 1.2M conformers (100 conformers per molecule), it takes on the orders of ~weeks to months to score each conformer for up to the 2,358 unique proteins for each evaluated method (evaluation time is 100x the time to score the ground truth data as we generate 100 conformers per molecule). As a result, we report the results for the first 100,000 conformer-protein interactions.

## L    LIMITATIONS

As demonstrated in §4.1-§4.4, CoarsenConf significantly improves the accuracy and reduces the overall data usage and runtime for conformer generation. However, CoarsenConf also has some limitations that we will discuss in this section.

**Autoregressive generation.**    While CoarsenConf improves accuracy with reduced training time and overall data, autoregressive generation is the main bottleneck in inference time. We linearize the input molecule based on spatially significant subgraphs and then process each one autoregressively. For a model with k torsion angles, we need $k + 1$ passes through our decoder. Coarse-graining is an effective strategy to reduce the number of decoder passes compared to traditional atom-wise autoregressive modeling. For example, for a given molecule, the number of torsion angles (which we use to coarse-grain) is significantly less than the number of atoms. Our choice of coarse-graining strategy allows us to break the problem into more manageable subunits, making autoregressive modeling a useful strategy, as it provides greater flexibility and control by allowing conditional dependence. CoarsenConf is a good example of the trade-offs that exist between generative flexibility and speed. We target this limitation by introducing a non-autoregressive version with an optimal transport loss. We see this improves the overall GEOM results, at the slight cost of a higher right tail of the error distribution.

**Optimal Transport.**    While CoarsenConf-OT trained with a non-autoregressive decoder with an optimal transport loss significantly outperforms prior methods and accomplishes the goal of effectively learning from traditional cheminformatics methods, the recall results still have room for improvement, especially for QM9. While CoarsenConf (no OT) achieves competitive results, we believe that continuing to focus on how to better integrate physics and cheminformatics into machine learning will be crucial for improving downstream performance. Due to the above concerns, we chose to evaluate our non-OT model on the property prediction and protein docking tasks, as wanted to use our best model denoted by the lowest overall RMSD error distribution.

**Approximate structure error.**    The success of learning the optimal distortion between low-energy and RDKit approximate structure depends on having reasonable approximations. While CoarsenConf relaxes the rigid local structure assumption of Torsional Diffusion in a way that leverages the torsional flexibility in molecular structures, it still depends on an approximate structure. This is a non-issue in some instances, as RDKit does well. In more experimental cases for larger systems, the RDKit errors may be too significant to overcome. We emphasize that the underlying framework of CoarsenConf is adjustable and can learn from scratch, not only the distortion from approximate RDKit structures. In some cases, this may be more appropriate if the approximations have particularly high error. We leave to future work to explore the balance between the approximation error and the inductive bias of learning from approximate structures, as well as methods to maintain flexibility while avoiding the issues of conditioning on out-of-distribution poor approximations.

**Equivariance.**    As CoarsenConf uses the EGNN (Satorras et al., 2021) framework as its equivariant backbone and thus only scalar and vector operations, there is no simple way to incorporate higher-order tensors. As the value of using higher-order tensors is still actively being explored, and in some

cases, the costs outweigh the benefits, we used simple scalar and vector operations and avoided expensive tensor products. We leave exploring the use of higher-order equivariant representations to future work, as it is still an ongoing research effort (Han et al., 2022).

