# OpenReview forum: "CoarsenConf: Equivariant Coarsening with Aggregated Attention for Molecular Conformer Generation"
_ICLR.cc/2024/Conference — Submitted to ICLR 2024_

### Official Review · Reviewer_NaAP · 2023-10-25

**Soundness:** 2 fair
**Presentation:** 2 fair
**Contribution:** 2 fair
**Rating:** 5
**Confidence:** 3

**Summary:**

The authors proposed a coarse-grained method for molecule conformer generation, through an SE(3)-equivariant hierarchical VAE. The method is able to do coarse-graining generation with variable length via an aggregated attention strategy. The proposed method achieved state-of-the-art performance across a set of downstream tasks, including structural precision, property prediction, and docking binding affinity.

**Strengths:**

The performance is promising.

**Weaknesses:**

1. Problem Significance: The authors may need to demonstrate the problem of conformation generation remains significant, in the context of the rapid development of 3D molecule generation from scratch.
2. Novelty: There has been a line of work studying coarse-grained molecule generation in the community [1, 2]. The authors may need to further discuss the novelty of their methods in comparison to these existing methods.
3. Novelty Again: There has been another work proposing its information fusion attention that is similar to the aggregated attention strategy [3].

[1]. Jin et al. Junction Tree Variational Autoencoder for Molecular Graph Generation. https://arxiv.org/pdf/1802.04364.pdf
[2]. Zhang et al. Molecule Generation For Target Protein Binding with Structural Motifs. https://openreview.net/forum?id=Rq13idF0F73
[3]. Wang et al. Retrieval-based Controllable Molecule Generation. https://arxiv.org/pdf/2208.11126.pdf

**Questions:**

N/A

---

> ### Author Response · Authors · 2023-11-15
>
> We are glad that you noticed that we achieve state-of-the-art performance in a number of important tasks, including structural precision, property prediction, and docking binding affinity.
>
> We want to clarify that all of the references brought up here have no direct overlap with our method or task, with most being examples of methods for 2D molecule generation, which is a completely different task. We address the comments specifically below.
>
> **> Question: Problem Significance of MCG**
>
> - Molecular Conformer Generation (MCG) remains a significant task, as many real-world drug development workflows employ lead optimization, which is a molecule-conditioned task by definition [1]. MCG is directly used for drug discovery and is often the first step to protein docking workflows such as AutoDock Vina and DiffDock [2] (both of which do not generate the ligand from scratch). To use any Vina-based software, a conformer must first be generated. It is more realistic to do this from a 2D graph, as generating molecules from scratch still possesses significant validity issues [3]. This would cause immediate failures to any off-the-shelf docking method, such as those which are currently used in industrial drug discovery.
> - Furthermore, **most methods that generate 3D molecules from scratch are 10-1000x times slower than CoarsenConf (CC)**, making it infeasible for many domain use cases [4]. The scratch-generation methods pay little attention to the underlying energetics of the generated structures, which is an extreme importance for chemists and domain scientists more generally.
> - Improvements to MCG are directly applicable to current polymer, catalyst, and nanoparticle design as it is a conditional task, whereas generating molecules from scratch cannot be as feasibly adapted due to the size and complexity of the domain shift [10].
> - It is also important to note that ML methods are only recently barely beating RDKit for conformer generation, which has been around since 2015 [9], so even beyond CC, more work must be done to generate accurate and low-energy conformers. Furthermore, we added additional evaluations that showcase CC can generate better structures for protein docking and energy prediction.
>
> **> Question: Novelty**
>
> - We explain the difference between the given references and our method in more detail below:
> - JT-VAE [5], uses a 2D pre-computed vocabulary, whereas we learn the fragments directly from their torsion angles for a completely different 3D task.  **Both methods are trying to accomplish different modeling goals and cannot be directly compared in any form of evaluation.** The only things these two methods have in common is that they are VAEs and utilize fragments differently, with [5] not being SE(3)-equivariant.
> - Zhang et al. [6] was designed for the structure-based drug discovery benchmarks. It relies on a fragment vocabulary (which we do not), and as a result, [6] cannot generalize to any molecule and is limited to those covered by the fixed vocabulary.
> CC can operate on any drug-like molecule, no matter the fragment breakdown, and can be extended to any coarsening criteria (see Appendix A).
>    - Furthermore, CC learns a latent representation of the CG molecule (structure, and atomic features) that can be used in both autoregressive and non-autoregressive generation techniques.
>   - [6] requires an iterative approach, where a motif binding site is selected and filled in by choosing one of the available fragments from its vocab. CoarsenConf also learns FG and CG positions, as well as distances and angles, leaving nothing fixed.
>   - We note we cannot compare with the SBDD benchmarks as presented in [5] directly, as they generate a molecule conditioned on the protein, whereas we target the off-the-shelf docking workflow where the ligand is known, and then must create the 3D structure.
>   - We are also the first to employ fragment-based generation in MCG.
>
> - RetMol [7] is a 2D task with embedding cross attention, whereas our Aggregated Attention specifically uses conditional 3D queries to predict the desired 3D shape and coordinates. Furthermore, our attention implementation operates over learned CG coordinates, not invariant features.
>   - Unlike to prior CG work, we utilize the entire learned representation, as discussed in Section 2. We note that information fusing is a central point of any cross-attention mechanism, and we use attention in a completely different way.
>   - We have additional novelty in how we leverage attention to enable variable-length coarse graining without a vocabulary, and to learn the FG positions from the entire latent space (which prior CG methods are unable to do). [7] is another example of another entirely different task, where the only overlap is the molecular domain and the use of attention. Our attention scheme and usage have never been used before, as the queries are constructed to yield the shape necessary for equivariant 3D construction.

---

> > ### Author Response · Authors · 2023-11-15
> > **Citations**
> >
> > [1] Rethinking drug design in the artificial
> > intelligence era https://discovery.ucl.ac.uk/id/eprint/10089268/3/Fisher%20RETHINK_manuscript_14Sep2019_final.pdf
> >
> > [2] Corso et al. DiffDock https://arxiv.org/abs/2210.01776
> >
> > [3] Rothchild et al. Investigating the Behavior of Diffusion Models for Accelerating Electronic Structure Calculations https://arxiv.org/pdf/2311.01491.pdf
> >
> > [4] Peng et al. Pocket2Mol: Efficient Molecular Sampling Based on 3D Protein Pockets https://arxiv.org/abs/2205.07249
> >
> > [5] Jin et al. JT-VAE. https://arxiv.org/pdf/1802.04364.pdf
> >
> > [6]. Zhang et al. https://openreview.net/forum?id=Rq13idF0F73
> >
> > [7]  Wang et al. RetMol: RETRIEVAL-BASED CONTROLLABLE MOLECULE GENERATION https://arxiv.org/pdf/2208.11126.pdf
> >
> > [9] RDKit ETKDG https://pubs.acs.org/doi/10.1021/acs.jcim.5b00654
> >
> > [10] Patra et al. Data-Driven Methods for Accelerating Polymer Design https://pubs.acs.org/doi/10.1021/acspolymersau.1c00035

---

> ### Comment · Reviewer_NaAP · 2023-11-22
> **Thank the authors for the rebuttal**
>
> The problem significance is well-illustrated by the authors. For novelty, although CoarsenConf is the first work to study coarse-graining in molecular conformation generation, and it is exciting to hear that the proposed method is not limited to a fixed vocabulary, the main body of coarse-graining workflow largely follows previous works, and the key idea behind the aggregated attention - utilizing a reference library to enhance the quality of generation - is not quite new. But overall, I am convinced that the authors have made some unique contributions, so I updated my rating to 5.

---

> > ### Author Response · Authors · 2023-11-23
> > **Thank you reviewer NaAP**
> >
> > Thank you for your response, and for recognizing that we have made unique contributions. We want to further clarify some things around your additional comments:
> >
> >
> > **> Main body of coarse-graining workflow and previous works**
> >
> > We want to emphasize that our work is quite different from many previous coarse-graining workflows. There is a long way to go to perfect the use of coarse-graining for molecular tasks and MCG in general, and it is a huge area of study [1]. This means that just because a paper does coarse-graining does not mean it is similar, as there are a huge range of approaches here. As we mentioned earlier, the prior works that you cited are very different, as
> >  - 1) they look at completely different problem settings, and
> >  - 2) have a very different coarse-graining procedure and setting (just including a coarse-graining in a paper doesn’t mean there’s that much overlap, as there are many nuances to doing this effectively).
> > Indeed, we believe that our paper is a contribution towards a new coarse-graining strategy to learn better representations, and has a different workflow from these prior papers.
> >
> > [1] Bottom-up Coarse-Graining: Principles and Perspectives https://pubs.acs.org/doi/full/10.1021/acs.jctc.2c00643
> >
> > **> Aggregated attention uniqueness**
> >
> > We emphasize that our aggregated attention mechanism accomplishes something that is entirely unique:
> > - 1) Our aggregated attention mechanism uses queries in the raw 3D coordinate space to learn molecular structures from the learned latent representation.
> > - 2) **Unlike CoarsenConf, all prior work limits all attention operations to the embedded feature space.**
> > - 3) **CoarsenConf does not use a reference library or fragment vocabulary to enhance the quality of generation**, as it operates directly on the input molecule. This is an important distinction, **as our method has more flexibility and can work on any given molecule**, not only those that can be parsed by the provided vocab.
> >  - 4) We set up conformer generation as a translation problem to go from RDKit approximation to low-energy structures (Sec. 3). As a result, we construct attention queries from the raw source language to the target language embedding. This is equivalent to having raw source text being fed into an attention-head of a large language model for machine translation. **This is something that has never been done**, and **cannot be done in prior methods**, as the text domain requires an embedding, or some form of text-to-numerical or byte encoding to enable matrix multiplication. This is important, as learned embeddings come at a cost of loss of information, and our method avoids this issue entirely. **Our Aggregated Attention mechanism leverages the inductive bias of our 3D translation problem to circumvent the need for encoding, and as a result, uniquely operates on raw molecule structures.**
> >
> > Overall, CoarsenConf is the first attention-based method to not require an embedding for the attention queries, as it mixes molecular structure space with a learned latent representation. We are also the first coarse-graining method to utilize attention to enable variable-length CG, compared to prior methods which force a fixed-length encoding.

---

### Official Review · Reviewer_qQEF · 2023-10-31

**Soundness:** 3 good
**Presentation:** 3 good
**Contribution:** 3 good
**Rating:** 6
**Confidence:** 3

**Summary:**

This paper proposes a molecular conformer generation framework based on the coarse-graining of molecular graphs. Its main idea is to learn a coarse-grained latent representation based on an encoder with a "multi-resolution" message passing structure and autoregressive ly decode the conformer from coarse latent representations. Experiments demonstrate performance improvement over existing works such as torsional diffusion for applications like property prediction and oracle-based protein docking.

**Strengths:**

Overall, I think this work provides a solid and incremental contribution to molecular conformer generation.

- To my knowledge, this work is the first to apply coarse-graining for molecular conformed generation.
- It is interesting to see that autoregressive decoding is still useful for molecular conformer generation (compared to existing diffusion-based techniques).
- The proposed idea can be extended to other tasks like generating molecules from scratch.
- The experiments seem solid enough to verify the usefulness of the proposed method.

**Weaknesses:**

Since the proposed architecture is a bit complex, it is hard to identify the main source of performance improvement in the architecture. For example, one might argue that most of the improvement comes from (a) using substructures with fixed 3D coordinates and (b) using an encoder with a pooling layer. However, (a) has been proposed by torsional diffusion paper, and (b) has been investigated by the GNN community. It would be nice if the authors could design an ablation study on the effectiveness of each architectural component.

**Questions:**

I have the impression that this paper is in fact quite related to the torsional diffusion paper, e.g., both paper uses molecular substructures as fixed building blocks for molecular conformed generation.

Could the authors elaborate more specifically on the difference and the benefits of using the coarse-graining procedure?

---

> ### Author Response · Authors · 2023-11-15
> **Addressing provided weaknesses**
>
> We are glad to see that the reviewer found our improvements for MCG novel. We respond to all comments below, and discuss a number of ablation studies that we conducted, as well as the differences between CoarsenConf and Torsional Diffusion.
>
> **> Question: Finding the source of improvement in the architecture**
>
> This is a very interesting question as the architecture is complex, and we have done our best to run key ablations.
> We agree that testing this hypothesis is quite challenging, but important, as removing the coarsening/pooling layers of our architecture would strip the model of our main inductive biases. An example of a conditional VAE for MCG with no pooling/torsion angle information, nor equivariant updates, is found in CVGAE [1], which does not perform well (but was a crucial building block for all MCG methods).
> **We conducted several ablations where we removed key components of our model to see if it was necessary for performance:**
> - 1)  We softened the definition of what a torsion angle is, as done in Torsional Diffusion. Specifically, we allowed for non-physical rotations like rotations around double and triple bonds. Using this coarsening strategy, CoarsenConf had poor conformer reconstruction, and training would not converge. Our method is sensitive to the coarsening strategy being physically plausible, which is crucial to this specific task of low-energy conformer generation. It is possible that a different coarse-grained strategy may prove useful for different modeling objectives.
> - 2) We also removed the distance-based auxiliary loss. This resulted in the interatomic distance error steadily growing during training resulting in infeasible sampled structures. Since unlike TD, CoarsenConf can update angles, coordinates, and distances at will, it is very important to correctly optimize all modalities. This is because if one modality is left uncontrolled, it tends to turn into a main source of error.
> - 3) Along with experimenting with both autoregressive and non-autoregressive architectures for 3D molecule tasks, our main focus was to introduce more evaluation methods that prior MCG methods did not evaluate on, which mimic real-world docking workflows. Due to the SBDD docking evaluations taking over 2 weeks per method, we did not do an extensive hyperparameter sweep, but provided an initial hyperparameter ablation based on RMSD in the appendix.
> - 4) To get closer to the question of what part of the architecture makes the most impact, we also ran a study on the amount of equivariant feature mixing in our decoder architecture (Eqn 10b). We found that increasing or decreasing the magnitude of mixing had little impact on the loss performance. From this, we believe that while the ability to break the problem down into fragments is beneficial, especially for downstream docking, there exist many equivalent versions of the architecture that differ with the weightings of the equivariant updates.

---

> > ### Author Response · Authors · 2023-11-15
> > **Answering questions**
> >
> > **> Differences with Torsional Diffusion**
> >
> > We have a number of differences with Torsional Diffusion (TD), which we highlight here:
> > - 1) We first clarify that while TD operates over fixed 3D coordinates, CoarsenConf does not. **Our method directly updates 3D coordinates and torsion angles directly, while also having influence over interatomic distances via the equivariant message passing updates**. We believe that by not restricting the method to one form of geometric updates, i.e. either pure angles or coordinates, we are able to achieve significantly better results, especially in downstream docking tasks.
> > - 2) Torsional Diffusion treats all coordinates and bond distances as fixed, making updates only to torsion angles. In contrast, CoarsenConf makes direct updates to coordinates, bond distances, and torsion angles, leaving every geometric modality open for control and optimization. Unlike TD, nothing in our method is ever used as a fixed building block, as the FG and CG coordinates and distances, as well as torsion angles, are all updated by the model.
> > - 3) CoarsenConf and TD use different definitions of torsion angles, as TD allows non-physical restorations around double and triple bonds (we do not, as explained in point 1) of the ablations, above). We also use torsion angles very differently. TD uses them as the entire modeling objective via diffusion, whereas CoarsenConf uses them to create a hierarchical learning framework to model FG and CG coordinates. We believe that having more degrees of geometric freedom can be beneficial, especially for downstream protein docking, as demonstrated in Table 3-4 and Fig. 5.
> > - 4) Another difference is that TD creates a layer of abstraction by not defining what a torsion angle is to the model. If a molecule has 8 torsion angles, TD outputs an 8-dim update vector, which is then applied over the torsion angles outside the model. In contrast, CoarsenConf operates directly over the true definition of torsion angles, as we coarsen along rotatable bonds, creating a hierarchical latent encoding that then controls the flow of information via message passing. By coarsening, we encourage our model to understand the angles it is trying to model, along with the individual atomic 3D positions.
> > - 5) As you mentioned, by learning a representation for the coordinates and atomic species, we can extend CoarsenConf to other generative tasks quite easily in ways Torsional Diffusion can not, due to its level of abstraction and model setup.
> >
> > **> Benefits of using coarse-graining**
> >
> > - Along with the advantages of coarse-graining that we describe above, one of the fundamental benefits of coarse-graining stems from their usage in 2D fragment-based molecule generation (detailed in Appendix A), as well as protein and molecular dynamics studies, which we introduce in Section 2. Coarse-graining simplifies the size of the system, and allows more efficient updates on the most crucial information. Without coarse-graining, autoregressive generation would be infeasible because of the large number of atoms, especially as we do not remove hydrogens. Furthermore, on a geometric level breaking down the problem into separate planar subspecies makes it significantly easier to model their rotational and positional interactions with nearby pieces.
> >
> > [1] CVGAE https://www.nature.com/articles/s41598-019-56773-5

---

### Official Review · Reviewer_vgnz · 2023-10-31

**Soundness:** 3 good
**Presentation:** 2 fair
**Contribution:** 2 fair
**Rating:** 5
**Confidence:** 3

**Summary:**

This paper proposes CoarsenConf, a novel conditional hierarchical VAE for molecular conformer generation. In particular, CoarsenConf aggregates the fine-grained atomic coordinates of subgraphs connected via rotatable bonds to create a variable-length coarse-grained latent representation, and uses a novel aggregated attention mechanism to restore fine-grained coordinates from the coarse-grained latent representation.

**Strengths:**

1. The idea of this work is straightforward and novel.
2. The entire model can be trained end-to-end, and generate more accurate conformer ensembles compared to prior generative models. Besides, it shows very good performance on multiple downstream applications.

**Weaknesses:**

1. Authors say that they are the first model to employ variable-length coarse-graining. As far as I know, it has already been used in the molecular field (e.g., Qiang B, Song Y, Xu M, et al. Coarse-to-fine: a hierarchical diffusion model for molecule generation in 3D, ICML2023).
2. The model architecture needs further explanations. For example, the description of encoder architecture in your appendix is unclear. What are the inputs and outputs of the three modules? How to get outputs based on inputs? Please reorganize this section.
3. The presentation needs further improvement. This paper does not provide any algorithm for the proposed method, making me very confused about a lot of training and inference details.
4. As shown in Table 1, Table 5 and Table 6, the performance of this work seems to be suboptimal, especially for Recall.

**Questions:**

1. Compared with current ML methods, how efficient is this method?
2. Why is there a lack of comparison with Geodiff in many experiments? They have already released their code.

---

> ### Author Response · Authors · 2023-11-15
>
> We are glad that you found the idea straightforward and novel, and pointed out that we had very good performance on multiple downstream applications. We answer all comments below, and have also made a number of updates to the paper to reflect our responses (including new algorithms to describe the encoder architecture, training, and inference), which are in red.
>
> **> Question: Variable-length Coarse-to-fine**
>
> - Thank you for bringing this paper to our attention! As it was published only 2 months (July ‘23 at ICML) before the ICLR deadline, we had not seen it at the time of the deadline. We have updated our paper to include this paper in Appendix A, and to clarify that we are the first to employ variable-length coarsening without a fragment vocabulary (as done in Qiang et al.) This is an important difference, **as our method has more flexibility and can work on any given molecule**, not only those that can be parsed by the provided vocab.
> When [1] and CoarsenConf are limited to the same atom types, if a new fragment not in the vocab were to be encountered at test time, [1] would have to be retrained from scratch whereas CoarsenConf can handle any molecular substructure.
>
> **>  Question: Encoder Architecture clarity**
>
> - We have updated our paper to add new algorithms and clarify the description of the encoder architecture in Appendix D, including what the inputs and outputs are of the three modules, and getting outputs based on inputs. The encoder is set up similarly to a bi-LSTM, but in our case, we have three equivariant message-passing modules that pass information from the fine module to the pooling module, to the final coarse-grain module in each layer. We bring up the analogy to the LSTM, as the product of one module is used as the input for the next.
> - At a high level, our encoder takes in (X, h) and (X^, h),  the ground truth and RDkit approximated conformer (equivariant 3D positions/features, invariant atomic features), and produces Z and Z^, the CG latent equivariant representation for the ground truth and RDKit inputs.
> - Each module is an EGNN update, which takes in equivariant features and invariant features, and produces a message-passing update for both, given by equations 6, 7, 8, and 9 for each of the provided modules. These equations detail the method of achieving outputs from inputs, but not the exact methodology for how the information is passed from one module to another, which we go into further detail next.
> - In more detail, as part of our preprocessing for each molecule, we initialize a fine-grain (FG), pooling, and coarse-grain (CG) graph scaffolds (correct shape zero init) with a coarsening strategy that we know a priori (in our case, by torsion angles). Thus, we know the n, n+N, and N nodes for each graph, along with their chemical bond and distance-based auxiliary edges during preprocessing. Then, we can initialize graphs to conduct the hierarchical message passing. We refer to these as the FG, pooling, and CG graphs in the appendix, which provide the edges, and nodes that are used in their respective equations. In each module, the respective graph features are updated and then used to pass information from one module to another. **This is shown in Algorithm 1 in Appendix D, which demonstrates the forward pass of a single batch.**
>
> **> Question: Algorithm for method (Training and Inference)**
>
> - We have added training and inference algorithms in Appendix D, which are labeled **Algorithm 2 and Algorithm 3**. To summarize, we encode the data into our coarse-grain latent representation, sample from the latent space, use Aggregated Attention to decode from CG to FG, and run our FG decoder to produce the desired structures.

---

> > ### Author Response · Authors · 2023-11-15
> >
> > **> Question: Sub-optimal results (Table 1, Table 5, Table 6)**
> >
> > - We note in Tables 1 and 5 that CoarsenConf achieves the best QM9 precision, including with a number of additional baselines added (Table 5 in Appendix).
> > - For DRUGS, CoarsenConf (CC) is very competitive with Torsional Diffusion (TD), and outperforms all other methods (as seen in Table 1). We are better than Torsional Diffusion on Median AMR, and are very close on the other GEOM-DRUGS metrics (for example, 52.0 to 52.1 on mean coverage). **Importantly, as seen in Figure 4, CoarsenConf has the lowest RMSD distribution of any model.** This means that in general, CC samples structures with lower error compared to all other models, including TD.
> > - Unlike Precision, Recall is extremely dependent on the chosen sampling budget, with performances degrading by over 50% when restricted to more feasible sampling budgets (i.e., less computationally expensive). As seen in Figure 7, CC is comparable, and better in Coverage, on Recall compared to all other models (including TD) when the sampling budget is lower (and more realistic for how sampling would be done in the real world).
> > - Additionally, we include multiple experiments that test the quality of our generated DRUGS structures that go beyond RMSD. Note that RMSD is not frequently used by domain scientists, so we also did comprehensive testing of the quality of our generated conformers on downstream, real-world applications:
> >   - 1) property prediction, where CoarsenConf generates conformers with the most accurate energy: **we reduce the $E_{min}$ error of TD by 50% (Table 2)**.
> >   - 2) Two downstream real-world applications, where our generated DRUGS structures are tested on how they would operate in practice for protein docking. CC outperforms all prior methods by a large margin, with **improvements of up to 53%** compared to the next best method (as you noted, in Tables 3 and 4, Figure 5).
> >
> > **> Question: Efficiency of CoarsenConf compared to other methods**
> >
> > - CoarsenConf was trained on 83% less training data than Torsional Diffusion and GeoMol via using 5 conformers per molecule, instead of 30. This resulted in equal or better performance with significantly less time and data usage.
> > We emphasize that CoarsenConf-OT was trained for 15 hours, compared to Torsional Diffusion and GeoMol which are 7-11+ days.
> > - There is not a large margin of difference between all tested methods at inference time. CoarsenConf-OT is faster than Torsional Diffusion, while base CoarsenConf with an autoregressive decoder is slightly slower.
> >
> > **> Question: Lack of comparison with GeoDiff**
> >
> > - As we noted in Appendix J, we tried very hard but were not able to run the GeoDiff code because of dependency conflicts with torch-scatter and torch-geometric. However, the TD authors shared their GeoDiff test set conformers for us to run our evaluation. Furthermore, unlike Torsional Diffusion which has built-in functionality to generate conformers from a custom set of SMILES, the public GeoDiff code only has the ability to re-run the RMSD evaluations (building out that functionality for GeoDiff would require new dataloaders, preprocessing, and evaluation scripts). For these reasons, we could not evaluate GeoDiff on the protein docking evaluations which take 2+ weeks per method.
> > - We initially provided the same baselines as in Torsional Diffusion, and have now added more model baselines from prior papers (including GraphDG, CGCF, ConfVAE, ConfGF, and DMCG), that were evaluated on a different random subset of GEOM molecules for RMSD evaluation. The paper has been updated to reflect this.
> > - One comment is that there is a train/test data mismatch issue, and it has been encountered before (more details can be found at [2]). Other methods, like DMCG (that do not compare to TD), use a different but similar test set, and also only publish their DRUGS results for a coverage threshold of 1.25, compared to our 0.75 angstroms (a higher coverage threshold gives better results). Given this, we left blank coverage results, which are a function of AMR, and AMR values can be roughly compared as both test sets are just different equal-sized subsets (see the updated Table 6).
> > - These additions further demonstrate that CoarsenConf outperforms many prior methods, while using less data and compute, and offers a large improvement on real-world downstream tasks.
> >
> > [1] Qiang B, Song Y, Xu M, et al. Coarse-to-fine: a hierarchical diffusion model for molecule generation in 3D, ICML2023
> >
> > [2] https://openreview.net/forum?id=w6fj2r62r_H&noteId=eTl4eNd2IDw

---

> > > ### Comment · Reviewer_vgnz · 2023-11-22
> > > **Thank you for your reply**
> > >
> > > Thank you for your reply. Most of my questions have been resolved, so I will raise my rating to 5.

---

> > > > ### Author Response · Authors · 2023-11-23
> > > > **Thank you Reviewer vgnz**
> > > >
> > > > Thank you for your response. Please let us know if you have any additional questions that you feel were not addressed.

---

### Official Review · Reviewer_6Hjw · 2023-11-02

**Soundness:** 3 good
**Presentation:** 3 good
**Contribution:** 3 good
**Rating:** 8
**Confidence:** 4

**Summary:**

This paper studies molecular conformer generation (MCG). The proposed method is a new SE(3)-equivariant hierarchical variational autoencoder that leverages coarse-grains molecular graphs with torsional angels. The proposed attention mechanism enables variable-length coarse-to-fine generation that restores high-quality conformers from coarse-grain graphs in an autoregressive way. This framework more efficiently generates more accurate conformer ensembles.

**Strengths:**

1. **Novelty.** As the authors claimed, this paper proposed a novel pipeline to generate conformers using a SE3-equivariant hierarchical VAE and aggregated attention.
2. **One stone three birds:** **efficiency, flexibilty and quality.** In contrary to prior works, the proposed method can generate all size of conformers by a single model whereas some existing approaches require a model for one length resulting in 100+ models to learn a dataset. This unified model learns more parameter-efficiently and more effectively with virtually more samples per model.
3. **Competitive performance.** Experimental results in Table 3 and 4 show that the proposed method achieve competitive performance on Protein Docking and Binding Affinity compared to two or three baselines.

**Weaknesses:**

1. Weak performance on GEOM-DRUGS compared to Torsional Diffusion. In addition, the performances of baselines are different from the literature. Also, Recall should be reported as well. Please explain what causes the discrepancy.
2. Only few baselines are provided. If more baselines are provided, then it will be better to evaluate the effectiveness of the proposed method compared to recent techniques.

**Questions:**

1. QM9 and ZINC250 have been used for learning molecular distributions. Also, several representations have been used for generation such as string (SMILES, SELFIES) and (2D) graphs. Is it possible to compare MCG with other graphs or string based methods? Often papers provide other groups of approaches in tables as references with dim fonts.
2. Coarse-to-fine is a popular strategy in many applications. The conditional generation idea can be generalized in other directions. 2D to 3D generation is quite popular in the computer vision domain. Have you ever considered other coarse representations? and how robust/sensitive is the proposed pipeline to the quality of coarse-grain generation?

---

> ### Author Response · Authors · 2023-11-15
> **Addressing provided weaknesses**
>
> We appreciated that the reviewer found our work to be novel, and noted that our work hits all three areas of efficiency, flexibility, and quality of the generated conformers. We answer all of your comments below, add more baselines, and have also made updates to the paper, which are in red.
>
> **> Question: Performance on GEOM-DRUGS**
>
> - We note that CoarsenConf (CC) is very competitive with Torsional Diffusion (TD), and outperforms all other methods (as seen in Table 1). We are better than Torsional Diffusion on Median AMR, and are very close on the other GEOM-DRUGS metrics (for example, 52.0 to 52.1 on mean coverage). **Importantly, as seen in Figure 4, CoarsenConf has the lowest RMSD distribution of any model.** This means that in general, CC samples structures with lower error compared to all other models, including TD.
> - Additionally, we include multiple experiments that test the quality of our generated DRUGS structures that go beyond RMSD. Note that RMSD is not frequently used by domain scientists, so we also did comprehensive testing of the quality of our generated conformers on downstream, real-world applications:
>   - 1) property prediction, where CoarsenConf generates conformers with the most accurate energy: we reduce the $E_{min}$ error of TD by 50% (Table 2).
>   - 2) Two downstream real-world applications, where our generated DRUGS structures are tested on how they would operate in practice for protein docking. CC outperforms all prior methods by a large margin, with improvements of up to 53% compared to the next best method (as you noted, in Tables 3 and 4, Figure 5).
>
> **> Question: Performance on baseline different from the literature.**
>
> - The numbers in our table are from generating molecules by running the different codebase models, following their public instructions. When available, we also used their published pre-trained weights. These numbers do have a slight discrepancy from the original papers. (We note that in running these baselines, some metrics, such as TD performance on AMR, are a little better than what was previously published in the paper).
>
> **> Question: Recall**
> - We publish recall values for all evaluated methods for 7 different sampling budgets in Appendix J, Figure 7. This issue of recall scores being very different based on the sampling budget was discussed in "Advantages and limitations of RMSD-based metrics” on page 7 (section 4.1).
> - Unlike Precision, Recall is extremely dependent on the chosen sampling budget, with performances degrading by over 50% when restricted to more feasible sampling budgets (i.e., less computationally expensive). As seen in Figure 7, CC is comparable, and better in Coverage, on Recall compared to all other models (including TD) when the sampling budget is lower (and more realistic for how sampling would be done in the real world).
>
> **> Question: Only a few baselines are provided**
>
> - We initially provided the same baselines as in Torsional Diffusion. We went and added more model baselines from prior papers (including GraphDG, CGCF, ConfVAE, ConfGF, and DMCG), that were evaluated on a different random subset of GEOM molecules for RMSD evaluation. The paper has been updated to reflect this. **CoarsenConf outperforms all of these additional baselines.**
> - One comment is that there is a train/test data mismatch issue, and it has been encountered before (more details can be found at: https://openreview.net/forum?id=w6fj2r62r_H&noteId=eTl4eNd2IDw). Other methods, like DMCG (that do not compare to TD), use a different but similar test set, and also only publish their DRUGS results for a coverage threshold of 1.25, compared to our 0.75 angstroms (a higher coverage threshold gives better results). Given this, we left blank coverage results, which are a function of AMR. AMR values can be roughly compared as both test sets are just different equal-sized subsets (see the updated Tables 5 and 6). As noted in the appendix, we tried very hard, but were not able to run the GeoDiff code for downstream applications because of dependency conflicts with torch-scatter and torch-geometric. However, the TD authors shared their DRUGS-generated conformers for GeoDiff evaluation.

---

> > ### Author Response · Authors · 2023-11-15
> > **Answering questions**
> >
> > **> Smiles Representation**
> > - It is not possible to compare MCG with string or SMILES-based approaches (that do not convert it to a 2D graph), as to our knowledge, none exist. The task of MCG is inherently graph-based, as for each node we need a 3D position, and these positions must respect the 2D connectivity (and 3D energetics). This is not found in a SMILES string, due to its issue of enumeration and canonicalization [1]. Many language model approaches rely on SMILES enumeration or other augmentation techniques, which would violate the equivariance of the MCG task, but it is an interesting future research direction.
> > - We do think that language model representations may be able to be used along with equivariant graph models in a similar way to EquiFold [4], but this has not been done for molecules, and it is an open research area.
> > - We note there is a method, DMCG [2] (which we provide as a baseline; CoarsenConf has better performance), that uses AlphaFold’s structure module to learn conformers from a pure 2D input, but uses over 100x the model parameters. It is also inefficient due to its loss function enumeration, which is done because of the removal of explicit equivariance in parts of the model architecture.
> >
> > **> Other coarse-grained representations**
> >
> > - This is a good point that there are multiple coarse-graining strategies, and indeed, the best way to coarse-grain molecules is an active research area in chemistry and physics. We coarsened along contiguous rotatable fragments separated by single bonds, i.e. torsion angles. However, we did an ablation and tested Torsional Diffusion’s torsion angles, which allows for non-physical rotations like rotations around double and triple bonds. Using this coarsening strategy, CoarsenConf had poor conformer reconstruction, and training would not converge. Our method is sensitive to the coarsening strategy being physically plausible, which is crucial to this specific task of low-energy conformer generation. It is possible that a different coarse-grained strategy may prove useful for different modeling objectives.
> >
> > **> How robust/sensitive is the proposed pipeline to the quality of coarse-grain generation**
> > - The quality of the latent representation was crucial to model performance. If there were any imbalance between the KL divergence in the learned prior and posterior, the model would not converge. As a result, we implemented self-targeting KL tuning. Unlike beta annealing where the KL weight gets larger, we set a target trade-off between the KL loss and reconstruction loss, and the model will adjust its weighting to achieve this goal [3]. By directly controlling the flow of information in our latent space, we created a robust latent representation that was relatively unphased by changes to hyperparameters in the underlying architecture.
> >
> > [1] SMILES enumeration https://pubs.acs.org/doi/10.1021/ci00057a005
> >
> > [2] DMCG https://arxiv.org/abs/2202.01356
> >
> > [3] Self Balancing KL https://arxiv.org/pdf/2103.06089.pdf
> >
> > [4] EquiFold https://www.biorxiv.org/content/10.1101/2022.10.07.511322v1

---

### Author Response · Authors · 2023-11-22

Dear reviewers,

As the discussion period comes to an end soon, we wanted to highlight the main changes and updates that we have made, which are also in our individual responses to each reviewer.

- 1) We demonstrate the best conformer accuracy via the lowest overall RMSD distribution (Fig 4). This means that in general, CoarsenConf samples structures with lower error compared to all other models, including Torsional Diffusion (TD). We also generate the best conformers as measured by energy calculations (Table 2), as well as in real-world conformer workflows such as rigid and flexible protein docking (Tables 3-4 and Fig 5 ), by a significant margin.
     - We note this is especially important as in practice, 3D molecular structures are primarily evaluated by their energy and binding affinity to targeted proteins (as opposed to RMSD). As a result, it was crucial to introduce large-scale ligand-centric docking evaluations to MCG.
- 2) We added additional GEOM baselines from 5 prior MCG methods. Here, CoarsenConf achieved state-of-the-art QM9 results and the best medium AMR for DRUGS when compared to all of these other prior methods. We also demonstrate the instability of the minimum RMSD Recall metric, and provide all results for all tested sampling budgets.
     - Unlike Precision, Recall is extremely dependent on the chosen sampling budget, with performances degrading by over 50% when restricted to more feasible sampling budgets (i.e., less computationally expensive). As seen in Figure 7, CoarsenConf is better on Coverage and is comparable on Recall when compared to all other models (including TD), when the sampling budget is lower (and more realistic for how sampling would be done in the real world).
- 3) We added encoder, training, and inference algorithms to our Appendix D.
- 4) We added additional ablations for CoarsenConf, to better understand which components are critical for stable performance:
     - Alternative coarse-grained structure for encoding → non physical flexibility prevents model convergence.
     - Sensitivity to KL Divergence in VAE paramterization → any KL imbalance results in failure to converge and as a result implemented self-adaptive KL regularization.
     - Auxiliary loss function ablations →  without stable interatomic distances the model fails to converge
     - Equivariant feature mixing in CG module of the encoder → important but exact scaling has minimal performance impact.


We believe that we have addressed all your comments here, as well as through a number of new baselines, ablations, and updates to our manuscript. We would be happy to answer any additional questions. Thank you!

---

### Meta-Review · Area_Chair_qdUn · 2023-12-06

**Metareview:**

This paper studies molecular conformer generation and proposes a new SE(3)-equivariant hierarchical variational autoencoder that leverages coarse-grained molecular graphs with torsional angles. The introduced model is then validated on several standard benchmarks with a comparison to a range of baselines.

The review scores are mixed, and we had a discussion during the reviewer-AC discussion period. The main concern is the empirical study, which most reviewers felt was inadequate, and recent baseline models such as GeoDiff are not compared. One reviewer further raised concerns about the novelty where similar ideas have been used in the past. Given those problems. I recommend rejection and hope the authors can resubmit this work to a future venue.

**Justification For Why Not Higher Score:**

See the metareview

**Justification For Why Not Lower Score:**

N/A

---

### Decision · Program_Chairs · 2024-01-16

Reject